

# Trends in the tropospheric general circulation from 1979 to 2022

Adrian J. Simmons

European Centre for Medium-Range Weather Forecasts, Shinfield Park, Reading RG2 9AX, UK

*Correspondence to*: adrian.simmons@ecmwf.int

**Abstract.** Atmospheric general circulation changes since 1979 are examined using the ERA5 reanalysis. Maps based on linear trends and time series for specific areas are presented. Attention is concentrated on monthly, seasonal and annual means, but shorter-timescale variability is also considered, including extremes. Changes in near-tropopause winds are the main focus, but related changes in temperature, wind and other variables throughout the troposphere are discussed.

Middle- and upper-tropospheric warming is larger in the subtropics and outer tropics than in the deep tropics, except over the Pacific. This is linked with a strengthening and meridional expansion of the tropical easterlies that has received little previous attention. Warming over several mid-latitude and subtropical land areas comes close to matching the large warming of the Arctic. Westerly upper-level winds in general weaken over the Arctic in winter, but strengthen in northern middle latitudes, contrary to arguments based on circulation changes due solely to amplified Arctic warming. The jet-stream region

over the eastern North Atlantic and western Europe shifts southward. Westerlies strengthen in a band stretching south-eastwards from the tropical western Pacific to southern Australia, and in the polar-jet-stream region that surrounds Antarctica.

Extreme jet-stream winds increase over the North Atlantic. Net kinetic energy also increases, mostly associated with sub-monthly variability along the mid-latitude storm tracks and over the tropical Pacific. Available potential energy changes less.

Geopotential height shows a distinct pattern of change in the stationary northern-hemispheric long-wave structure. There are increases in surface pressure over the North Pacific and southern mid-latitudes, and decreases over the Arctic Ocean and offshore of Antarctica.

Several comparisons are made between ERA5 and the JRA-55 reanalysis, and between ERA5 and the observations it assimilated. They show reassuring agreement, but some regional differences require further investigation.

## 1 Introduction

Warming of the Earth's climate system has been deemed unequivocal in each of the last three assessment reports of the Intergovernmental Panel on Climate Change, most recently in the Sixth Assessment Report (AR6; IPCC, 2022). Multidecadal tropospheric temperature change nevertheless varies substantially with geographical location, height and



season. The implications for atmospheric circulation of the amplified near-surface warming of the Arctic (Serreze and Barry, 2011) have been the subject of quite considerable discussion, but it has increasingly been realised that several other factors are in play. Among them are the warming of regions such as south-eastern Europe and the Middle East at rates similar in magnitude to those observed over much of the Arctic, a generally larger warming over land than sea, a weakening of Arctic amplification with increasing height and a relatively large rate of warming of the upper troposphere at subtropical and outer-

tropical latitudes. AR6 has provided a timely discussion of much of this.

Changes in upper-tropospheric winds are linked to changes in surface flow and horizontal temperature gradients through the tendency of the atmosphere to remain close to thermal-wind balance. Where hydrostatic and geostrophic balance apply, the vertical shear of the wind is proportional to the temperature gradient across the direction of flow. The proportionality factor is larger at low than high latitudes. Changes in monthly or seasonally averaged winds are in general not simply related to

external forcings of temperature change, however, as there are typically accompanying changes to the transports of heat and momentum by transient disturbances. The various tendencies for change are reconciled by the divergent secondary circulations that preserve balance. Hoskins and Woollings (2015) provide an account of various processes that may be involved in changes in circulation regimes.

Uncertainty regarding changes in circulation stems from the natural variability of the atmosphere and from differences in the

observationally based datasets that are used to study change. These in turn cause the results of studies to be sensitive to the choice of datasets, periods and metrics of change. Moreover, it can be difficult to attain a holistic view of change as many studies focus on a specific constituent circulation system or feature rather than the general circulation as a whole.

Gulev et al. (2022) concluded in AR6 that it was "likely" that several aspects of the large-scale tropospheric circulation had changed since the mid-20th century. Despite reporting progress, only an increase in strength of the Walker Circulation since

1980 was judged to be "very likely". In addition, "high confidence" was expressed in a trend of the Southern Annular Mode (SAM) to become systematically more positive, particularly in austral summer. Many of the studies that were assessed were based on reanalysis data, but the European product used tended to be ERA-Interim (Dee et al., 2011) rather than the newer ERA5 (Hersbach et al., 2020; Bell et al., 2021).

These considerations motivate a new look at the tropospheric trends depicted by ERA5. What is termed ERA5 here uses the

ERA5.1 update for 2000-2006, as this gives a better representation of the upper troposphere and lower stratosphere (Simmons et al., 2020). Linear trends and confidence intervals have been derived for the period March 1979 to February 2022, either from data for all months or from three-month seasonal averages for March to May (MAM), June to August (JJA), September to November (SON) or December to February (DJF). The quality of ERA5 and other reanalyses for this 43-year period benefits from the observing systems put in place for the Global Atmospheric Research Programme's 1979

global observing experiment, systems which have subsequently been sustained and refined. The period is also that during which global warming has been most pronounced and for which human influence on this warming has now also been termed unequivocal (IPCC, 2022).



Some comparisons with results from JRA-55 (Kobayashi et al., 2015) are reported. JRA-55 is one of the more recent reanalyses and the most suitable alternative to ERA5 for present purposes. It was used along with ERA-Interim, MERRA-2 (Gelaro et al., 2017) and some older reanalyses in a number of the studies assessed in AR6 and referenced here. The multi-decadal consistency of the fits of the ERA5 data assimilation to several types of wind observation provides further confidence in the main results presented for near-tropopause winds. Consistency with aircraft data over the North Atlantic is also reported for a case of extreme jet-stream strength.

The plan of this paper is as follows. The next section covers quite familiar ground, updating estimates of trends in surface-air temperature and introducing the calculation of confidence intervals. Section 3 discusses temperature trends in the free troposphere. Section 4 presents the core results of the study, relating to the trends in monthly and seasonally averaged near-tropopause winds. Instantaneous jet-stream wind maxima are the subject of Sect. 5. Sections 6 and 7 discuss changes in lower-tropospheric wind and in geopotential height and surface pressure. Section 8 examines trends in kinetic and other forms of atmospheric energy. It is followed by a concluding discussion.

## 2 Surface-air temperature

### 2.1 Global-mean trends

Figure 1(a) presents time series from March 1979 to February 2022 of anomalies in monthly and globally averaged two-metre temperatures from ERA5. Anomalies are relative to averages for each month of the year made over the 43-year period, but are plotted as deviations from the ordinary least squares linear trend for the period. Black dots show averages over the World Meteorological Organization's standard climatological reference periods of 1981-2010 and 1991-2020. They lie close to the linear trend line. Deviations from this trend line (the so-called regression residuals) are primarily due to the natural variability of the atmosphere about the changing climatic state. Among them are the warm periods associated with the 1997/98 and 2015/16 El Niño events, the cold spell following the eruption of Mount Pinatubo in 1991 and the longer "hiatus" period of predominantly below-trend temperatures from 2007 to 2014. The regression residuals also include contributions from the uncertainty of the ERA5 reanalysis and from the (also uncertain) deviation of actual climate change from the linear trend.

Confidence intervals for the rate of temperature change have been calculated following Santer et al. (2008). They depend on the variance of the regression residuals and use an effective sample size that is reduced (from 516 in the present case) depending on the one-month-lagged correlation of the monthly residuals. The same approach was used by Gulev et al. (2022) in their contribution to AR6, although their quoted 90 % confidence intervals are narrower than the 95 % confidence intervals that are generally reported here.

The global trend shown in Fig. 1(a) is 0.190 K/decade, with 95 % confidence interval ±0.026 K/decade. The one-month-lagged correlation of the regression residuals is 0.72 giving an effective sample size of 83, and the "standard error" of the linear fit to the data is .013 K/decade. The corresponding 90 % confidence interval is ±0.022 K/decade. The difference



between the 1991-2020 and 1981-2010 climatological means is 0.193 K, very similar to the change over a decade given by the least-squares fit.

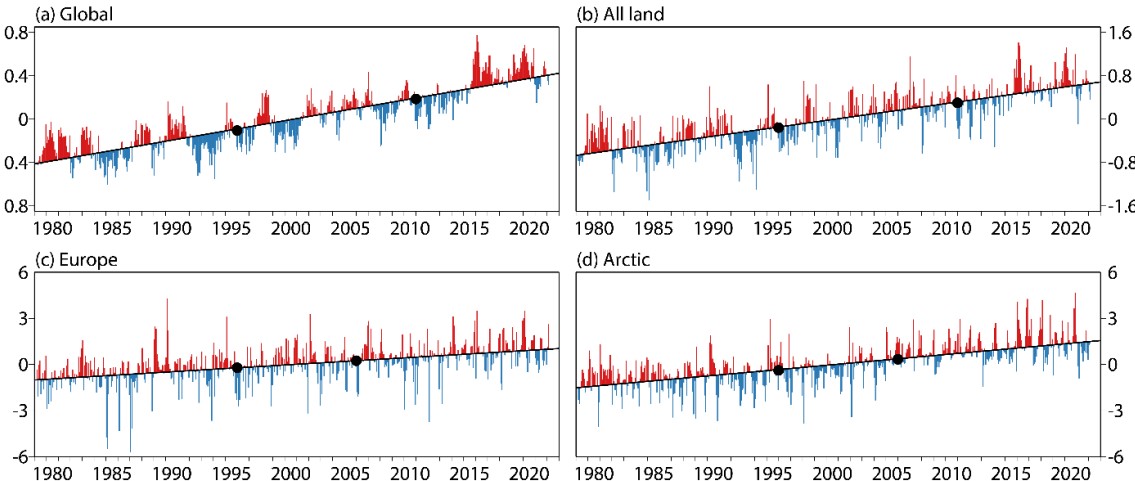

**Figure 1: Monthly means of ERA5 two-metre temperature (K) averaged (a) globally, (b) over all land, (c) over Europe (defined as land from 20° W-40° E and 35° N-80° N) and (d) over the Arctic (defined as the whole area north of 66.6° N). Values are plotted**
**from March 1979 to February 2022. They are shown relative to averages for each month of the year made over the period, and plotted according to whether they are above (red) or below (blue) the least-squares-fit linear trend for the period. The black circles plotted at the ends of 1995 and 2005 show thirty-year average values for 1981-2010 and 1991-2020.**

The linear trend increases to 0.240±0.042 K/decade when the period over which it is calculated is reduced to the latest thirty-year climatological reference period, 1991-2020. Although some of the increase may be due to an increased rate of climate
change, the trend for this thirty-year reference period appears to be particularly affected by natural variability, with a cool spell due to the Pinatubo eruption early in the period, and warm spells in 2015/16 and 2019/20 late in the period. Further illustration is given in Sect. 8 for components of atmospheric energy.

Warming has been larger globally over land than sea. Figure 1(b) is as Fig. 1(a) but for land areas only. The difference in vertical scale should be noted. The trend over land is about 60 % larger than the trend over the globe as a whole: 0.31
K/decade, with 95 % confidence interval ±0.03 K/decade.

Figure 1(c) shows the corresponding plot for the average over all land from 20° W to 40° E and 35° N to 80° N, which is labelled as Europe. It will be seen below that both the trend and the variability about the trend (as measured by the confidence interval) are larger for Europe than for any other continent, a reflection of Europe's relatively small size and location at the end of the North Atlantic storm track. The trend for Europe with 95 % confidence interval is 0.46 ±0.10
K/decade. The corresponding trend for the Arctic (Fig. 1(d)) is larger still, 0.70 ±0.10 K/decade.





## 2.2 Geographical variations

Figure 2 presents maps of the geographical variation of the linear trends from ERA5 and JRA-55, and provides information related to confidence. The globally complete ERA5 trend is shown in panel (a) and the JRA-55 trend in panel (e). Panel (b) shows the ERA5 trend only for those regions where the 95 % confidence interval is one-signed, indicating high confidence

that there has been warming (or in a few places cooling) over the 43-year period. Panel (b) picks out mostly regions of warming greater than 0.1 K/decade, but identifies cooling of larger magnitude than this around Antarctica. It should be noted that absence of a significant trend does not necessary imply low confidence in the computed trend. If there is a high level of confidence in a warming trend in one region and a cooling trend in a neighbouring region, then there is also high confidence that the trend is small somewhere between the two regions. A temperature trend that is locally small enough to be termed

"non-significant" because of the applied statistical test may nevertheless be significant if it is part of a regional temperature pattern that involves a substantial temperature gradient, as this in turn indicates wind shear through the thermal wind relationship. Shepherd (2021) discusses more comprehensively and generally the danger of reliance on statistical tools at the expense of physical reasoning and prior knowledge in climate science, whilst conceding that such tools are nevertheless useful heuristics.

ERA5 and JRA-55 depict a similar pattern of large-scale change, which in turn is similar to that shown for 1981-2020 from the HadCRUT5 dataset (Morice et al., 2020) in Fig. 2.11 of Gulev et al. (2022). Similar patterns can also be seen in the 1979-2018 trends presented by Simmons et al. (2021) for a larger number of datasets. As such, the principal features are already familiar. Warming occurs over most of the globe. It is largest in the Arctic, especially in a band from Svalbard eastward to beyond Novaya Zemlya, and is relatively high over eastern Europe, the Middle East, North Africa, parts of

eastern Asia and southwestern North America. There is cooling around the coast of Antarctica and over the eastern Pacific to the west of Chile and Peru. Features that are more minor but noteworthy as they will be seen to occur through much of the depth of the troposphere include the lower rates of warming (or indeed cooling) over western Canada and over the North Atlantic between Greenland and south-western Europe.

The standard deviation of the regression residuals shown (for ERA5) in Fig. 2(c) indicates primarily where natural variability

is large or small. Variability is large over Siberia, the higher latitudes of North America and the Antarctic. Aside from oceanic regions close to coasts or that are prone to variability associated with sea-ice cover, the largest variability over sea occurs in the equatorial eastern Pacific, where sea-surface temperature (SST) variability associated with El Niño and La Niña events is large. JRA-55 (not shown) has similar features, the most apparent difference being smaller variability over the Antarctic plateau.

The other factor influencing confidence intervals is the correlation of the regression residuals from one month to the next. Correlations are shown for ERA and JRA-55 in panels (d) and (f) of Fig. 2. Persistence of the largest temperature anomalies from one month to the next tends to be low in extratropical latitudes. One exception is the northern Barents Sea, which has the largest warming trend anywhere and high variability, but correlations larger than in neighbouring regions. The largest

correlations are nevertheless found over the tropical oceans, in particular over the eastern Pacific and to a lesser extent the

Atlantic.

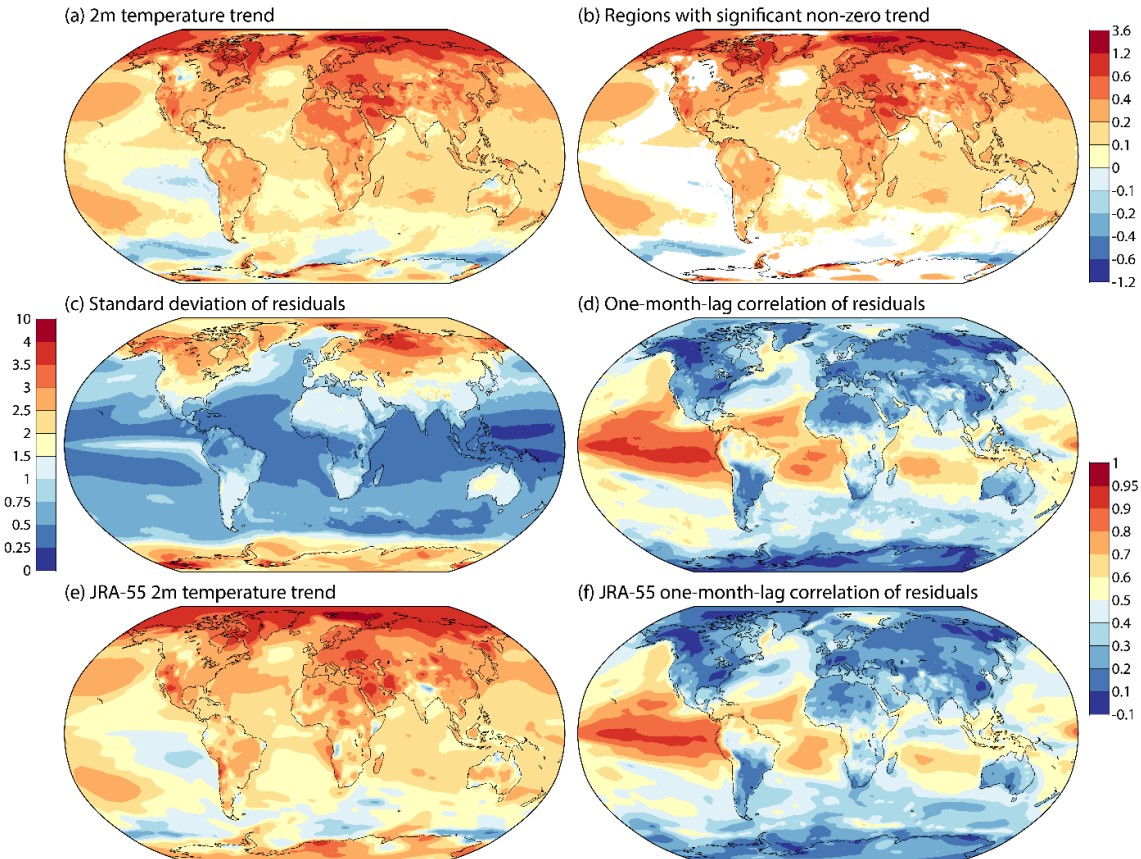

**Figure 2: (a) The linear trend (K/decade) in two-metre temperature based on monthly average ERA5 data from March 1979 to February 2022. (b) As (a), but with trends shown only where the 95 % confidence interval is one-signed. (c) The standard deviation of differences between two-metre temperatures and their linear trend (K). (d) The correlation coefficient between**
**successive monthly deviations of two-metre temperature from the linear trend. Panels (e) and (f) are as (a) and (d), but for JRA-55.**

Correlations of surface-air temperature are relatively low over the Gulf Stream for both ERA5 and JRA-55. They are low because of the variability both of sea surface temperature (SST) and of airflow over the Gulf Stream. Related to this, the trend in surface-air temperature in this region is smaller than the trend in SST, the opposite of the case for the average over

all ice-free sea. The latter is one of the sources of small differences between the global trends in surface-air temperature from reanalyses and the trends from monthly temperature datasets such as HadCRUT5 and GISTEMP (Lenssen et al., 2019) that use surface-air temperature over land but SST instead of marine air temperature (Simmons et al., 2017).

There is nevertheless good overall agreement among the various datasets. This is important because the calculations of confidence intervals do not account for systematic drifts in the errors of the individual datasets to which they apply.



Comparisons of multiple datasets have been reported by Simmons et al. (2021), but did not include confidence intervals. The updated sample presented in Table 1 shows how these confidence intervals vary little among datasets, which give a common picture of trends and uncertainty from one continental region to another. It includes the background forecasts of the ERA5 data assimilation system, an indicator of the performance of ERA5's assimilating model and use of data in general, since the background forecasts have only limited dependence on direct observations of surface-air temperature, as discussed by Simmons et al. (2004) for the ERA-40 reanalysis.

|  | Global | Europe | Asia | N Amer | Africa | Australia | S Amer | Arctic |
|---|---|---|---|---|---|---|---|---|
| **ERA5 an** | 0.19±0.03 | 0.46±0.10 | 0.36±0.07 | 0.32±0.08 | 0.33±0.04 | 0.13±0.08 | 0.23±0.04 | 0.70±0.10 |
| **ERA5 bg** | 0.18±0.03 | 0.45±0.10 | 0.35±0.07 | 0.31±0.08 | 0.29±0.04 | 0.18±0.08 | 0.20±0.04 | 0.70±0.10 |
| **JRA-55 an** | 0.18±0.02 | 0.46±0.10 | 0.35±0.07 | 0.32±0.08 | 0.25±0.04 | 0.12±0.08 | 0.19±0.04 | 0.74±0.10 |
| **GISTEMP** | 0.19±0.02 | 0.48±0.10 | 0.37±0.07 | 0.31±0.09 | 0.28±0.04 | 0.18±0.07 | 0.23±0.03 | 0.74±0.10 |
| **HadCRUT5** | 0.19±0.02 | 0.44±0.10 | 0.37±0.07 | 0.32±0.08 | 0.28±0.04 | 0.18±0.06 | 0.19±0.03 | 0.72±0.09 |

**Table 1: Least squares linear trends in monthly mean surface-air temperatures (K/decade) and the range of the 95 % confidence intervals for six continental averages for the period from March 1979 to February 2022 for the ERA5 analysis (an) and background (bg), and three other datasets. The continental areas are defined as all land within the following ranges:  Europe (20° W - 40° E, 35° N - 80° N), Asia (60° E - 180° E, 0° - 85° N), North America (170° W - 50° W, 15° N - 85° N), Africa (25° W - 55° E, 40° S - 35° N), Australia (110° E - 160° E, 50° S - 10° S) and South America (90° W - 25° W, 65° S - 15° N). Global and Arctic averages are also shown; for GISTEMP and HadCRUT5.0.1.0 (downloaded 22 March 2022) they are based on sea-surface temperature not air temperature over ice-free sea. [Note: HadCRUT5 values are provisional, based on the calendar years from 1979 to 2021 pending availability of data for January and February 2022]**

The global trends have the lowest uncertainty, but confidence intervals are not much larger for trends over Africa and South America. The trends themselves differ quite highly among datasets for these two regions, but are generally larger for Africa than South America. As noted earlier, Europe has the largest trends, but also the largest uncertainty. This is consistent with the finding by Hawkins et al. (2020) that the largest signal-to-noise ratios for temperature increases are in tropical regions.

The trends over Australia from ERA5 and JRA-55 are lower than those from GISTEMP and HadCRUT5 shown in Table 1, and also lower than those from other such global datasets and the national ACORN dataset (Trewin, 2013). The ERA5 background is more in line with these other datasets, and the same is true for JRA-55 (not shown). Simmons et al. (2021) discuss several problematic aspects of the performance of ERA5 in analysing surface-air temperature over Australia.

Some of the local differences between ERA5 and JRA-55 or other datasets such as HadCRUT5 stem from differences in resolution. They are partly inherent to the various datasets, whose native resolutions vary from around 31 km for ERA5 to 5° of latitude and longitude for HadCRUT5. They also depend on processing choices made for this study, as the JRA-55 analyses were downloaded at 1.25° resolution whereas ERA5 fields have been processed with 0.25° resolution. Other local differences may stem from production issues. Two of those discussed by Simmons et al. (2021) for ERA5 are a spurious cooling to the north of Greenland, most evident in the March-May average shown in Fig. 3, and excessive warming over Iran south of the Caspian Sea. Both are associated with issues in the analyses for the 1980s: the specification of sea-ice



concentration north of Greenland and the combination of a cold background-model bias and limited availability of surface-
air observations over Iran.

## 2.3 Seasonal variations

Most of the regional features seen in the full-year trend maps shown in Fig. 2 are also seen in the seasonal trend maps
presented in Fig. 3, though there are differences. There is a strong seasonal cycle in the rate of temperature increase at high
northern latitudes, where summertime change over the Arctic Ocean is small because near-surface temperatures are
constrained to be close to 0° C by melting ice, a process known to be represented reasonably well in ERA reanalyses
(Simmons and Poli, 2015; Simmons et al., 2021). In addition, temperature increases in JJA over Arctic land are not
appreciably larger than the increases in several mid-latitude and sub-tropical regions. A large part of Siberia shows either
cooling or only weak warming in winter, but strong warming in spring, when feedback from a tendency for snow to melt
earlier comes into play. Warming is more pronounced in spring and summer than in autumn and winter over the western
USA, but higher in autumn and winter over the subtropical North Atlantic.

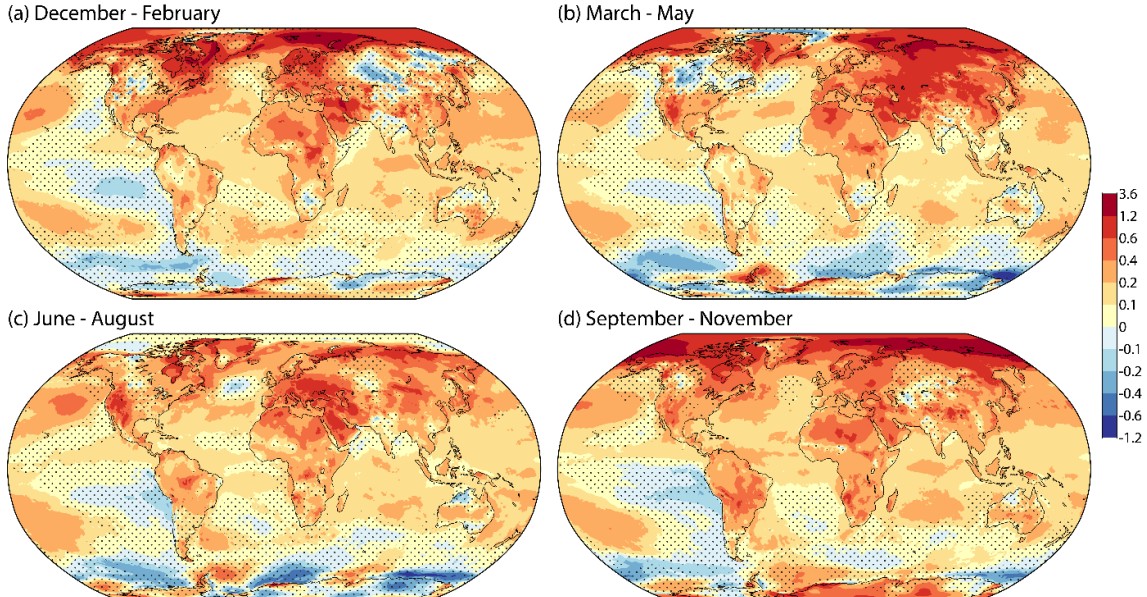

**Figure 3: The linear trend (K/decade) in seasonal-average ERA5 two-metre temperatures from March 1979 to February 2022 for
(a) DJF, (b) MAM, (c) JJA and (d) SON. Dots show where the 95 % confidence interval is two-signed.**

Cold-season cooling (or weakened warming) over North America and Asia has been linked with an increasing frequency of a
particular type of disruption of the stratospheric polar vortex (Cohen et al., 2021), although other factors are also in play
(Blackport et al., 2022). The winter trend for warming over East Asia as well as the Barents and Kara seas (where the
warming is associated with declining sea-ice cover) differs from the natural teleconnection pattern reported by Kim et al.
(2022), who showed from de-trended surface-air temperature anomalies that warm conditions over the Barents and Kara seas



tend to occur in conjunction with a circulation pattern associated with cold conditions over East Asia. Aside from such
specifics, interannual variability generally causes lower confidence in the trends over northern mid-latitude land for winter
than for any other season.

## 3 Upper-air temperature

Figure 4 presents maps of the ERA5 and JRA-55 trends in temperature at four standard pressure levels from 850 to 300 hPa.
Area averages and confidence intervals are shown in Table 2. The two reanalyses are again in broad agreement, although
there are some regional differences, more so lower in the atmosphere. There is net warming at each level. The rate of
warming is highest in the Arctic at 850 hPa and to a lesser degree 700 hPa, but highest at middle or subtropical latitudes in
the upper troposphere, where it peaks at close to the 300 hPa level shown.

Several regional trends extend from the surface to the middle or upper troposphere. They include the relatively low warming
rate of the region between southern Greenland and western Europe, which implies increasing westerly winds to the south and
decreasing winds to the north. A band of either cooling or relatively weak warming extends from the tropical central Pacific
south-eastwards to central Chile.  A region of cooling or only weak warming also occurs above the Antarctic coastline, with
generally stronger warming to the north of it, though ERA5 and JRA-55 differ in detail here. Further discussion of these and
other regions is given in the following section, where near-tropopause wind changes are considered.

|  |  | Global | Europe | Asia | N Amer | Africa | Australia | S Amer | Arctic |
|---|---|---|---|---|---|---|---|---|---|
| **300 hPa** | **ERA5** | 0.18±0.05 | 0.25±0.06 | 0.20±0.04 | 0.23±0.05 | 0.23±0.06 | 0.24±0.07 | 0.17±0.07 | 0.17±0.12 |
|  | **JRA-55** | 0.17±0.05 | 0.24±0.06 | 0.17±0.04 | 0.24±0.05 | 0.23±0.06 | 0.21±0.07 | 0.21±0.07 | 0.15±0.12 |
| **500 hPa** | **ERA5** | 0.16±0.04 | 0.27±0.07 | 0.17±0.04 | 0.18±0.05 | 0.14±0.05 | 0.19±0.06 | 0.13±0.04 | 0.25±0.06 |
|  | **JRA-55** | 0.17±0.04 | 0.27±0.07 | 0.14±0.04 | 0.24±0.05 | 0.15±0.05 | 0.21±0.06 | 0.16±0.05 | 0.27±0.07 |
| **700 hPa** | **ERA5** | 0.16±0.04 | 0.29±0.08 | 0.21±0.05 | 0.15±0.06 | 0.18±0.05 | 0.12±0.06 | 0.11±0.04 | 0.32±0.07 |
|  | **JRA-55** | 0.19±0.04 | 0.31±0.07 | 0.19±0.05 | 0.25±0.06 | 0.17±0.05 | 0.20±0.06 | 0.12±0.04 | 0.39±0.07 |
| **850 hPa** | **ERA5** | 0.15± 0.03 | 0.36± 0.08 | 0.26±0.06 | 0.17±0.07 | 0.24±0.04 | 0.17±0.08 | 0.15±0.03 | 0.46±0.08 |
|  | **JRA-55** | 0.18±0.03 | 0.36±0.08 | 0.25±0.06 | 0.30±0.08 | 0.20±0.05 | 0.21±0.08 | 0.12±0.03 | 0.48±0.08 |

**Table 2: Least squares linear trends in monthly mean temperatures (K/decade) and the ranges of 95 % confidence intervals for six**
**continental averages (as in Table 1) for the period from March 1979 to February 2022, for four pressure levels and the ERA5 and**
**JRA-55 analyses. Global and Arctic averages are also shown.**

The most obvious of the differences between reanalyses seen in the maps is the isolated cooling over southern Africa at 850
hPa in JRA-55, but the largest differences in the continental averages occur in the lower to middle troposphere over North
America. Here JRA-55 exhibits a much higher rate of warming than ERA5, more akin to that seen for surface air from both
reanalyses. ERA-Interim also has a higher rate of warming than ERA5 over North America. ERA-Interim warms over the
continent at an average rate of 0.20±0.11 K/decade at 850 hPa and 0.23±0.08 K/decade at 700 hPa, from March 1979 to





February 2019. ERA5 also has a markedly lower rate of warming than JRA-55 over Australia at 700 hPa. In this case ERA-Interim's warming rate is lower still.



**Figure 4: The linear trend (K/decade) in monthly average ERA5 (left) and JRA-55 (right) temperatures from March 1979 to February 2022 for (a, b) 300 hPa, (c, d) 500 hPa, (e, f) 700 hPa and (g, h) 850 hPa. Dots show where the 95 % confidence interval is two-signed. Regions where the ERA5 model orography exceeds 1500 m (for 850 hPa) or 3000 m (for 700 hPa) are masked in white.**





The differences in temperature trends over North America and Australia are not a particular concern for this study as they are

of quite broad scale, and the more-local horizontal gradients of temperature are less different, consistent with both ERA5 and JRA-55 drawing closely to the wind data from the regions. Differences in the trends of near-tropopause winds will be seen in the following section to be small. Investigation for North America points to an issue with the performance of ERA5's variational bias adjustment of increasing numbers of temperature measurements from ascending and descending aircraft. ERA-Interim did not adjust for the biases of these data and JRA-55 did not use aircraft temperature data because of the

biases. Another factor is ERA5's use of a radiosonde temperature bias correction scheme (RICH; Haimberger et al., 2012) different to that used (albeit in different versions) by JRA-55 and ERA-Interim (RAOBCORE; Haimberger, 2007; Haimberger et al., 2008). The differences over Australia can be seen in panels (e) and (f) of Fig. 4 to extend from the Indian Ocean to New Zealand and over the Maritime Continent, raising the possibility of an issue with the bias correction of satellite sounding data. Background forecasts and analyses are colder than Australian radiosonde data around 700 hPa for

later but not earlier years.

Maps of trends of seasonal averages at 700 and 300 hPa from ERA5 are presented in Fig. 5. Corresponding plots for JRA-55 (not shown) exhibit generally similar inter-seasonal differences. Differences between ERA5 and JRA-55 over North America at 700 hPa are relatively small for MAM, but present in all seasons. JRA-55 has more extensive and stronger cooling than ERA5 at 300 hPa over East Antarctica and Siberia in all seasons.

A region of cooling stretches from central Canada to the central North Atlantic in the MAM average shown for 700 hPa. The same is found for trends at 850 hPa and 500 hPa. A related increase in westerly flow along the south of this band will be seen in the following section. To the north, the rate of warming of the Arctic decreases with increasing height in autumn, winter and spring, but increases in summer. Warming of middle and low latitudes at 300 hPa is largest in boreal autumn and winter. Warming is smaller in the deep tropics than the subtropics throughout the year, apart from over much of the Pacific Ocean.

## 4 Near-tropopause wind

### 4.1 Climatological state

Synoptic maps for the upper troposphere typically exhibit elongated jet-stream flows at extratropical latitudes.  Flows are predominantly from west to east, but often with meanders where meridional winds are large. Examples for extreme cases are presented in Sect. 5. The meanders vary in position, however, and little of them remains in climatological averages, whether

for year, season or month. Maps of time-averaged flow such as presented in Fig. 6 show that the bands of high zonal wind mostly have some degree of poleward tilt in the downstream direction, but this is associated mainly with a longitudinal variation in the latitude of maximum zonal flow. The tilts, whether in the zonal wind component or in windspeed, do not in general indicate the direction of the time-averaged flow.

Studies focused explicitly on long-term changes in upper-tropospheric jet streams may be based on a characterization of jet

structures, as was done by Pena-Ortiz et al. (2013) and Manney and Hegglin (2018), for example. This approach has some



advantages over examination of the time-averaged flow, especially regarding the characteristics of polar jet streams in the northern hemisphere. Interest here is not solely in the jet streams, however. Results are presented in terms of annual or seasonal averages, and the variations about these averages.

**Figure 5: The linear trend (K/decade) in 700 hPa (left) and 300 hPa (right) temperature based on seasonal-average ERA5 data from March 1979 to February 2022 for (a, b) DJF, (c, d) MAM, (e, f) JJA and (g, h) SON. Dots show where the 95 % confidence interval is two-signed. 700 hPa regions where the ERA5 model orography exceeds 3000 m are masked in white.**



The present approach is largely complementary and confirmatory to that based on jet-stream characteristics: the seasonal-mean wind structures shown in panels (c) to (f) of Fig. 6 are similar to the seasonal jet frequency distributions shown for 1980-2014 from the MERRA-2 reanalysis in the upper left panels of Figs. 1 to 4 of Manney and Hegglin (2018). Manney and Hegglin's study predates ERA5, but includes JRA-55 among the five reanalyses it considers. It provides a comprehensive account of jet-stream changes over the period studied, most of which are seen directly or can be inferred from results presented here from ERA5 for a somewhat longer period. In discussing a subset of these changes below, broad agreement with the findings of Manney and Hegglin's study should be taken as read unless stated otherwise.

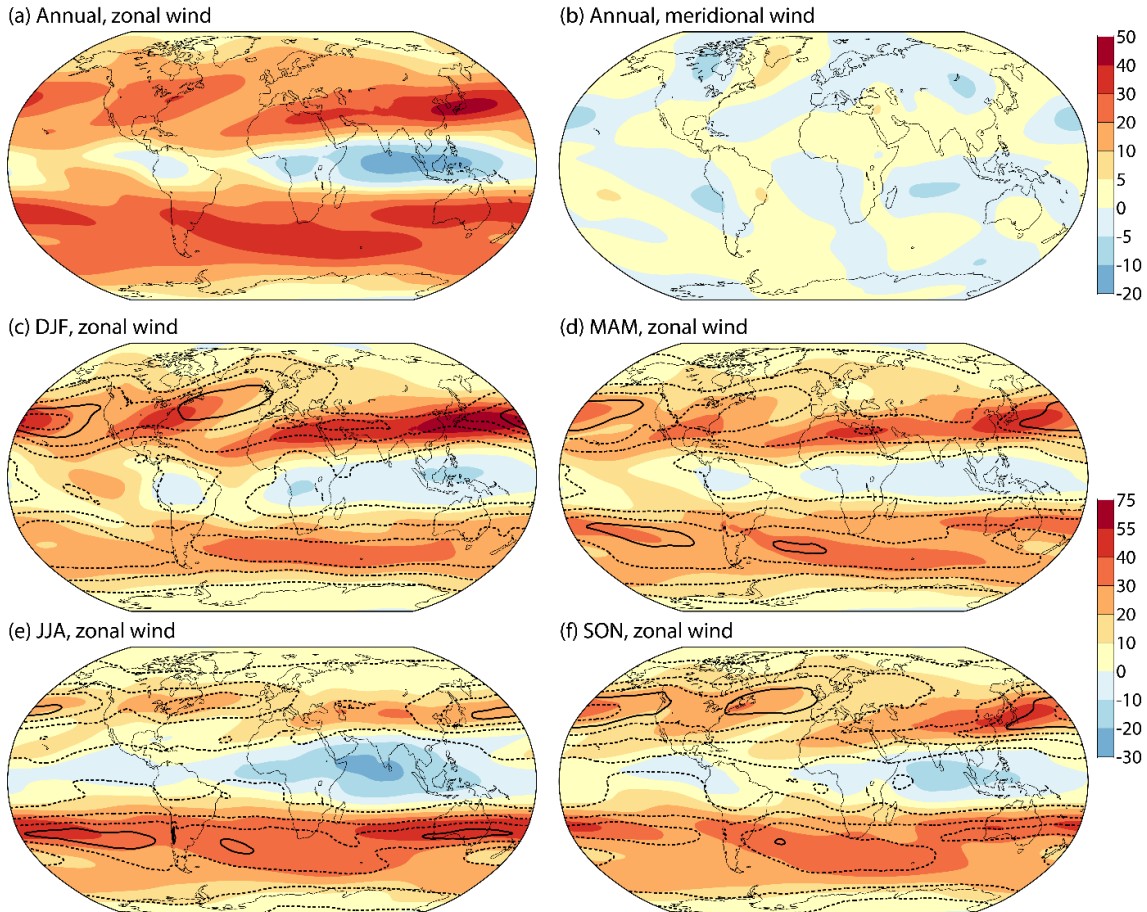

**Figure 6: ERA5 200 hPa wind (ms⁻¹) averaged from March 1979 to February 2022 for (a) the zonal and (b) the meridional component. Seasonal average zonal components for this period are shown (with different contour intervals) in panels (c) DJF, (d) MAM, (e) JJA and (f) SON. Black contours show the total sub-seasonal variation of the 200 hPa wind vector, with contour interval 160m²s⁻². Solid contours denote the 600m²s⁻² values.**

Panels (a) and (b) of Fig. 6 show climatological zonal and meridional components of 200 hPa wind derived by averaging all ERA5 analyses from March 1979 to February 2022. The annual-average meridional wind component is generally less than 5 ms⁻¹ in magnitude in the jet-stream regions where the annual-average zonal wind component is upwards of 30 ms⁻¹. The



most prominent meridional flow is in the trough over eastern Canada and Greenland, where meridional winds exceed 5 ms$^{-1}$ in the annual average and 10 ms$^{-1}$ in the winter average (not shown).

The corresponding seasonal averages of the zonal wind component are shown in panels (c) to (f), with different shading bands. Black contour lines denote the total sub-seasonal variation, the sum of the variances of the zonal and meridional wind components. The two variances are of similar magnitude.

The predominantly zonal time-averaged 200 hPa flow exhibits considerable variation with longitude as well as latitude, limiting what can be inferred reliably from changes in zonal averages. The most pronounced jet-stream region, evident in all
seasons and with the strongest peak winds, stretches from North Africa to the Pacific Ocean and beyond. For most of the year the average winds are strongest over or close to Japan, from where observations made at Tateno by Wasaburo Ooishi led to discovery of the jet stream in the 1920s (Lewis, 2003). Depending on season, the central latitude of this jet-stream region extends from 20-30º N over the north-western coast of Africa to 45-55º N over the western coast of North America.

The other main northern hemispheric jet-stream region is centred over the eastern seaboard of North America in all seasons
other than spring, although it will be seen below that the latter has been changing. This jet-stream region is shorter in zonal extent, is located further north and has a larger south-west to north-east tilt.

Hoskins and Hodges (2019a, b) have documented the climatology of northern hemisphere storm tracks for the upper and lower troposphere, based on ERA-Interim. Consistent with their findings, regions of large sub-seasonal variability can be seen in Fig. 6 to be aligned with the main jet-stream regions. Peak magnitudes occur downstream and a little equatorward of
the strongest time-mean flow, consistent with the baroclinic instability of the regions and their potential to trap and guide stationary Rossby waves (Hoskins and Woollings, 2015). Winds in the Arctic are largest and most variable over the seas around Svalbard, where warming of the lower troposphere has been particularly large. Although synoptic maps commonly show polar jet streams over North America and Eurasia, these jets tend to be relatively weak, variable in position, intermittent and strongest nearer to 300 hPa than 200 hPa. They thus do not feature clearly in the averages shown in Fig. 6,
although a hint can be seen in panel (d), which shows stronger springtime flow over northern than central Europe.

The subtropical jet-stream region that extends across Australia and much of the South Pacific has a larger seasonal variation in strength than its northern counterpart. A second region of high zonal wind runs from South America to the southern Indian Ocean. In all seasons (though weakly in DJF) the time-averaged flow is indicative of a double-jet structure extending from Australia across the South Pacific, with a minimum in average zonal flow located over or downstream of New Zealand.
Variability in the southern hemisphere is in general closely tied to the regions of high mean wind.

Easterly winds predominate at 200 hPa in the tropics. They are particularly strong over southern India and the Indian Ocean in JJA, the season when the strongest average northern hemispheric westerlies occur to the north-east. These easterly and westerly maxima are linked through the upper-tropospheric Tibetan Anticyclone. Easterlies encircle the globe in this season. In contrast, the 200 hPa tropical flow is westerly over the Pacific Ocean east of the International Date Line in DJF and
MAM.





## 4.2 Trends

Figure 7 shows rates of change of the zonal wind component, derived from least squares fits over all months. Complete global maps of the trends are presented in the left-hand panels and maps showing the trends only where the 95 % confidence interval is one-signed are presented in the right-hand panels.




**Figure 7: The linear trend (ms⁻¹/decade) in monthly average zonal winds from March 1979 to February 2022 for (a, b) ERA5 at 200 hPa, (c, d) ERA5 at PV=2, (e, f) JRA-55 at 200hPa and (g, h) the ERA5 background at 200 hPa. The right-hand panels show trends masked to show only where the 95 % confidence interval is one-signed.**





Looking first at the plots for the ERA5 analyses at 200 hPa shown in the uppermost panels, the most prominent feature in
terms of magnitude and statistical significance is a trend for strengthening of the tropical easterlies. This trend stretches
westward from the Maritime Continent to the eastern Pacific. It shows up clearly in the 1979-2018 seasonal trends of the
zonal average zonal wind from ERA5 shown in Fig 2.18 of Gulev et al. (2022), although the magnitude and significance of it
is masked in DJF by a strengthening of the westerlies over the tropical Pacific. Further discussion of these and other changes
is given below, after discussing the other panels of Fig. 7.

Panels (c) and (d) show corresponding trends for the dynamic tropopause defined by a potential vorticity of $2x10^{-6}$ $m^2$ $s^{-1}$K
$kg^{-1}$ (PV=2). A number of other results presented below for the 200 hPa level have also been compared with those for the
PV=2 surface. This was to look for sensitivity to differences between winds at 200 hPa and those at the tropopause, or
sensitivity to changes over time in tropopause height. As illustrated by the example in Fig. 7, only small differences were
found in the extratropics. Such differences as are seen for high latitudes in Fig. 7 appear to be related to the lower tropopause
at these latitudes, as the trends there for the 300 hPa pressure level (not shown) are closer to those for PV=2. The pressure of
the PV=2 surface varies from a little below 100 hPa in the deep tropics to a little above 300 hPa at the poles in the
climatological mean.

Differences are more substantial in the tropics. Here the trend in speed of the easterlies is less pronounced for PV=2. This is
because the increase in easterly wind speed is weaker at the high tropical tropopause than at 200 hPa. Checks for all ERA5
pressure levels from 150 to 300 hPa show little sensitivity to the choice of level, but 200 hPa provides a good compromise
between the strength of the climatological flow and the long-term rate of change of this flow.

Panels (e) and (f) of Fig. 7 are for 200 hPa, but from JRA-55 rather than ERA5. The two reanalyses are in good overall
agreement as regards changes in the tropics, and more so the extratropics. Local differences include a patch over southern
Nigeria where there is a decreasing trend in easterlies in JRA-55, in contrast to increasing easterlies in ERA5 that are more
consistent with what is analysed for the surrounding region. This difference is likely due to a different treatment of the wind
data from a single radiosonde station. JRA-55 has a larger rate of reduction in the speed of the westerly zonal wind
component over Antarctica, but the two reanalyses provide a similar picture of acceleration of the westerlies over and to the
immediate north of the Southern Ocean.

Panels (g) and (h) are also for 200hPa, and show the trends of the background forecasts from the ERA5 data assimilation.
The evident similarity between the trends for the background and the analyses is important, as it shows that the analysed
trends are not the consequence of a significant change over time in the extent to which biased background forecasts are
corrected by observations whose number, spatial distribution or type likewise changes over time. Further discussion is given
in Sect. 4.4 for key regions.

The statistical test indicates high confidence in the signs of only the largest rates of change, but the spatial coherence of the
trends, the consistency with patterns of temperature change, and the overall similarity between ERA5 and JRA-55 lend
credence to the patterns of wind change.



Figure 8 shows that the tropical easterlies strengthen in all seasons. The trend is consistent through thermal-wind balance with the trend shown earlier for temperature to increase faster in the subtropics than the deep tropics, except over the Pacific east of the Date Line. Despite extending around much of the equatorial belt, this trend received little if any discussion as

such in AR6, in which most of the attention on atmospheric circulation was focused on regional features. Among these is the Pacific Walker Circulation (see Sect. 2.3.1.4.1 of Gulev et al., 2022), for which Ma and Zhou (2016) have presented robust evidence of a strengthening and westward shift from 1979 to 2012. This was based on reanalysis datasets that included JRA-55 and ERA-Interim, which were in particularly close agreement. When characterized by the divergent component of the zonal wind, the upper-tropospheric flow of the dominant cells of the Walker Circulation comprise westerlies over the

tropical Pacific Ocean starting around 150° E, and easterlies that extend westward over the Indian Ocean but no further. The full wind field presented here on average changes from easterly to westerly at the Date Line. The strengthening and shift of these Walker Circulation cells can be seen nevertheless in Fig. 7, in the trends for stronger Pacific westerlies east of the Date Line, stronger easterlies over the Indian Ocean, and weaker easterlies between 150° E and 180° E. Figure 8 shows that the strengthening of the Pacific cell is largest in DJF.

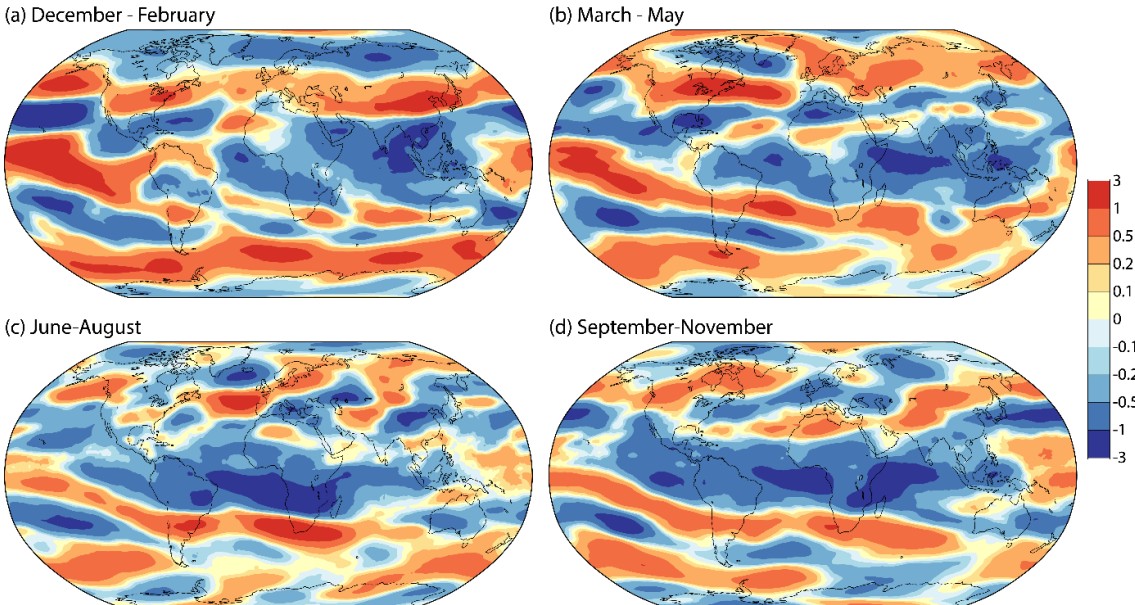


**Figure 8: The linear trend (ms⁻¹/decade) in 200 hPa ERA5 seasonal-average zonal winds for (a) DJF, (b) MAM, (c) JJA and (d) SON, from MAM 1979 to DJF 2021/22.**

Discussion of expansion of the tropical regime of the atmosphere has tended to focus on a widening and accompanying strengthening of the Hadley Circulation since the 1980s (see again Sect. 2.3.1.4.1 of Gulev et al., 2022; also Pikovnik et al.,

2022), but expansion and strengthening can also be discussed for the easterly upper-tropospheric flow. The latitudinal extent of the tropical and subtropical zone in which there is a negative trend in zonal wind is larger than the latitudinal extent of the tropical easterlies, implying that the area over which the wind is easterly has expanded. This has been quantified by





calculating the area between latitudes 30° N and 30° S where the zonal wind component is easterly, expressing it as a percentage of the total area between these latitudes. The calculation was made using hourly ERA5 analyses, but values were averaged monthly prior to assessment. The area of easterlies typically varies seasonally from around 20 % in April to 50 % in August at 200 hPa, though with interannual variability. The corresponding variation in strength is from about 7 to 12 ms$^{-1}$. Figure 9 shows monthly anomalies of the area and average strength of the easterly wind component. The percentage of the atmosphere between 30° N and 30° S where the 200 hPa wind has an easterly component increases at a rate of 0.67 percentage points per decade, with a 95 % confidence interval of ±0.33 %/decade. The trend is appreciably lower at 150 hPa (0.45±0.35 %/decade) and 500 hPa (0.48±0.30 %/decade), and somewhat larger at 300 hPa (0.74±0.35 %/decade). The rate of increase in strength of the average easterly wind component is largest at 200 hPa: 0.23±0.06 ms$^{-1}$/decade.

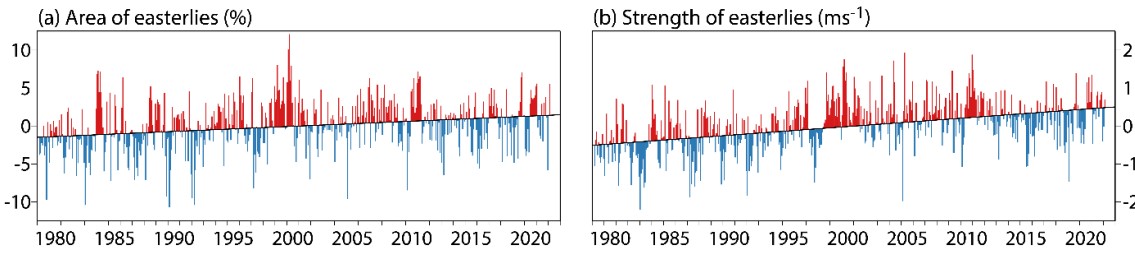

**Figure 9: Monthly means of (a) the area with an easterly component of 200 hPa wind between latitudes 30° N and 30° S as a percentage of the total area between these latitudes, and (b) the strength (ms$^{-1}$) of the easterly component of the wind averaged over this area. Values are shown relative to the average for each month in the period from March 1979 to February 2022, and plotted according to whether they are above (red) or below (blue) the least-squares-fit linear trend for this period.**

The statistical significance of the increase in westerly 200 hPa winds over the equatorial central and eastern Pacific is weaker. The average strength between 10° N and 10° S and 180° W and 85° W increases by 0.39 ms$^{-1}$/decade with 95 % confidence interval ±0.44 ms$^{-1}$/decade. The 90 % confidence interval is ±0.37 ms$^{-1}$/decade.

The trends in zonal wind at extratropical latitudes have a distinct banded structure. Over all months (Fig. 7), westerlies tend to weaken over the Arctic, but strengthen along a zone to the south. The strengthening is largest in DJF (Fig. 8), when the transition from weakening to strengthening westerlies occurs mostly between 50° and 60° N, and in MAM, when there is a predominant strengthening of westerlies at high latitudes over the Eurasian sector, with narrow bands of weakening and strengthening westerlies to the south in lower middle latitudes and the northern subtropics. The pattern of change is more fragmentary over the northern extratropics in JJA and SON.

Westerlies strengthen over two bands at most longitudes of the southern hemisphere, with a band of weakening westerlies in between that stretches eastward from the Pacific to at least the Atlantic. Some seasonal differences can be seen in Fig. 8, but they are generally smaller than for the northern extratropics.

The banded spatial structure of the trends makes it difficult to assess implications for the strength and location of the strongest winds, as most of the zones of strongest flow coincide with zones where change is small: changes tend to be more



in the meridional shear of the flow than in the peak flow. Comment is made here for three regions; further discussion is given in the following sub-section.

Firstly, the trend over all months is relatively weak in the vicinity of Japan, with at most only low significance attached to a small reduction in westerly flow in the region of strongest average wind. Secondly, westerly flow weakens over and to the east of southern Greenland and strengthens west of the Iberian Peninsula, consistent with the minimum in tropospheric warming over the Atlantic between Greenland and western Europe. This corresponds to a southward shift of the latitude of strongest mean flow over the eastern North Atlantic. Thirdly, westerly winds predominantly increase in the polar jet-stream zone surrounding Antarctica, particularly in DJF.

Although the meridional component of the wind is generally weaker than the zonal component in climatological averages, the long-term trends of the two components are of similar magnitude in the extratropics. This is not shown explicitly here, but can be inferred from the trends in geopotential height illustrated in Sect.7. Changes in wind speed are discussed below.

## 4.3 Nominal 43-year changes in wind speed

Discussion of changes in the extratropical 200 hPa flow is continued here in terms of the differences between nominal wind-speed averages for the beginning and end of the 43-year study period. These are derived from the 43-year average by adding and subtracting the change over 21.5 years provided by the least squares linear fit. The upper two pairs of maps in Fig. 10 show results for the speed of the monthly average wind fields from ERA5 and JRA-55. The two reanalyses are in close agreement, as indeed they also are for the seasonal averages shown later for ERA5 but not JRA-55.

The maps in panels (e) and (f) of Fig. 10 are based instead on monthly averages of instantaneous wind speeds from ERA5. Values are larger in this case, but patterns are similar, reflecting the similarity between the patterns of the speed of the monthly average wind and of the total sub-monthly variation. This similarity is much as shown earlier for the patterns of seasonal average zonal wind and the total sub-seasonal variation. Further discussion of nominal 43-year changes such as these are thus confined to results for the speed of the monthly average of ERA5 winds.

The changes shown in Fig. 10 are mostly small compared with the mean state, the largest exception being the change in speed of the tropical easterlies discussed earlier. Among the evident extratropical features is a strengthening of the wind in the region of strongest flow over the eastern seaboard of North America, a shift northward of the jet-stream region over the USA and a shift southward downstream over the North Atlantic and western Europe. Changes in strength occur along the subtropical jet over North Africa and Asia, with some strengthening and a northward shift of the region of strongest flow over the central and eastern North Pacific. There is a strengthening and equatorward shift of the region of strongest flow over South America, a weakening of the subtropical jet over and downstream of Australia, and a strengthening of the flow in the polar jet-stream region further south.



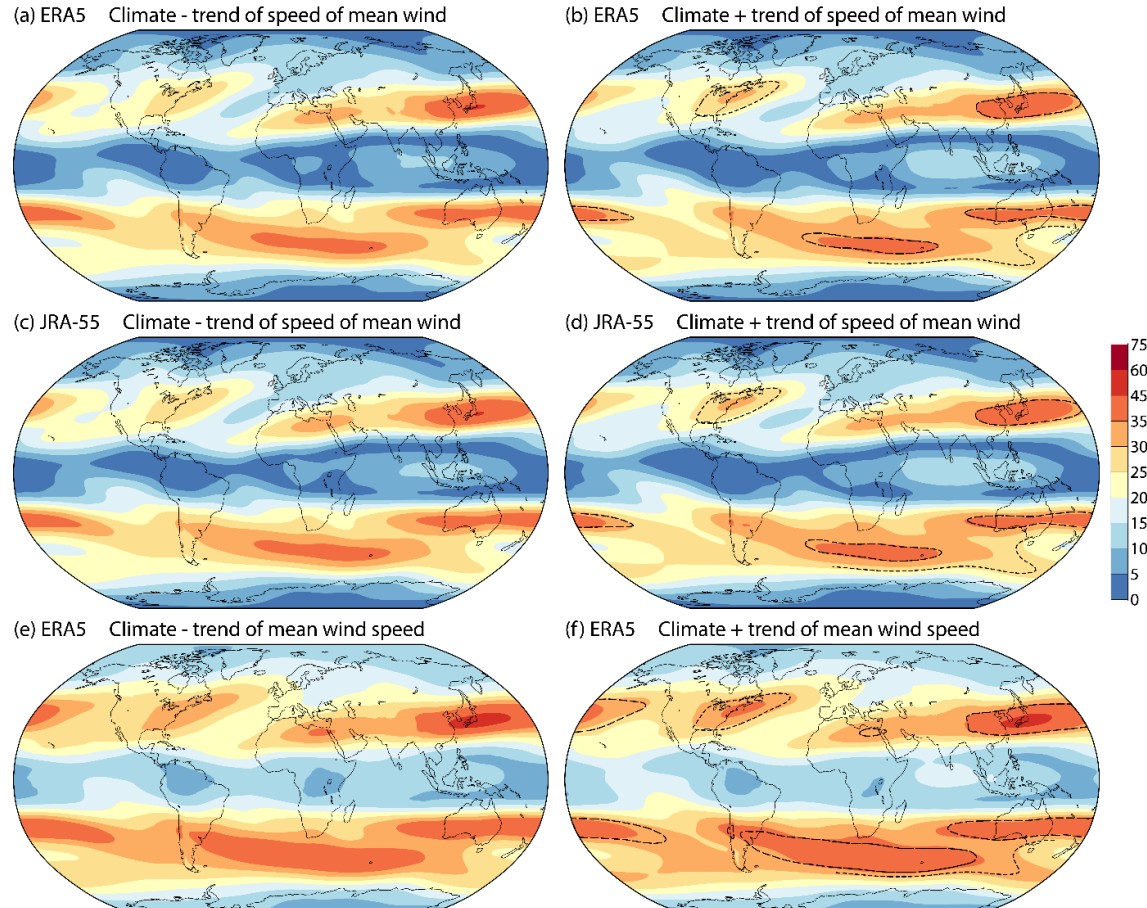

**Figure 10: 200 hPa wind speeds (ms$^{-1}$) based on subtracting (left) and adding (right) the change over 21.5 years derived from least squares linear fits of monthly mean wind components from March 1979 to February 2022 to the average wind components for the period, shown for (a, b) ERA5 and (c, d) JRA-55. Black dashed contours in panels (b) and (d) show the 35 ms$^{-1}$ contours from panels (a) and (c) respectively. Parts of the 25 ms$^{-1}$ contour are likewise transposed using shorter dashes. Panels (e) and (f) show corresponding ERA5 plots based on monthly means of hourly wind speeds. The shorter dashed black contours are for 30 ms$^{-1}$ contours in this case.**

These and other changes are quantified in Table 3. It indicates a slowing and small poleward shift of the strong jet stream close to Japan. In addition to the relatively large equatorward shift of the flow maximum over South America, there is a small equatorward shift south of Australia and New Zealand. Elsewhere, the main jet regions of the southern hemisphere shift slightly poleward, though by less than 1° of latitude at almost all longitudes.





| | | 0° | 30°E | 60°E | 90°E | 120°E | 150°E | 180° | 150°W | 120°W | 90°W | 60°W | 30°W |
|---|---|---|---|---|---|---|---|---|---|---|---|---|---|
| **N. America and N. Atlantic** | **speed (ms⁻¹)** | | | | | | | | | -1.5 | 1.5 | 1.4 | 1.3 |
| | **latitude (degrees)** | | | | | | | | | 1.6 | 2.8 | 0.9 | -1.5 |
| **Africa, Asia and N. Pacific** | **speed (ms⁻¹)** | 0.4 | -0.7 | 1.1 | 0.7 | -0.5 | -1.5 | -2.0 | 0.8 | 2.1 | | | 0.0 |
| | **latitude (degrees)** | -0.2 | 0.2 | -0.1 | 1.8 | 0.1 | 0.4 | 1.2 | 2.1 | 1.0 | | | 1.3 |
| **Subtropical Indian Ocean, Australia and S. Pacific** | **speed (ms⁻¹)** | | | | -0.3 | -0.8 | -1.0 | -1.8 | -3.9 | -2.8 | | | |
| | **latitude (degrees)** | | | | -0.5 | -0.7 | -0.3 | -0.1 | -0.6 | -1.3 | | | |
| **Extratropical S. America, S. Atlantic, Indian Ocean and S. Pacific** | **speed (ms⁻¹)** | -0.9 | 0.3 | 0.1 | 0.2 | 1.2 | 2.0 | 2.9 | 3.2 | | | 2.3 | -1.0 |
| | **latitude (degrees)** | -0.9 | -0.7 | -0.4 | -0.6 | -0.5 | 0.4 | 0.4 | -0.1 | | | 2.0 | 3.0 |

Table 3: Nominal changes in 200 hPa wind speed maxima from March 1979 to February 2022 for selected longitudes and four jet-stream regions, based on the fields shown in panels (a) and (b) of Fig. 10. Corresponding changes in the latitudes of the wind maxima are also shown.

The nominal 43-year changes in the speeds of seasonal-mean winds are shown in Fig. 11. Several changes identified above for annual averages can be seen to occur for all seasons of the year, though weakly in some. They include the strengthening of winds in the tropics and in the polar jet-stream region eastward from south of Australia.

Other changes differ quite markedly from season to season. Weakening of the strong mean flow close to Japan is most prominent in SON and northward movement of the jet-stream region in the eastern North Pacific is pronounced only in DJF. The maximum of the mean flow near the eastern seaboard of North America increases in DJF and JJA, but decreases in SON. The region of strongest flow narrows in DJF. A more radical change is seen for MAM, where the decline of the flow maximum over the Gulf Coast of the USA is consistent with a trend throughout the troposphere for stronger springtime warming over the continental US land mass than over the Gulf of Mexico. Southward movement of the region of strongest mean wind over the eastern North Atlantic and western Europe is pronounced in DJF and JJA, weaker in MAM and absent in SON.

It should be noted that the SON changes for the northern extratropics may not indicate a fundamental weakening of the autumnal circulation, as they could instead indicate a lengthening of the natural summer season that is not taken into account when fixed calendar months are used to designate seasons.

The strengthening of winds over and to the south of South Africa is most prominent for JJA, and poleward movement of the jet-stream region over the South Atlantic in DJF is counter-balanced in the annual mean by equatorward movement in other





seasons. The nominal 43-year movement in the annual mean wind maximum over the South Atlantic is poleward only from 18° W eastward, and does not exceed 1° of latitude. Equatorward movement exceeds 1° of latitude west of 22° W, and peaks at 4.6° of latitude at 45° W.


**Figure 11: Seasonal 200 hPa wind speeds (ms⁻¹) based on subtracting (left) and adding (right) the change over 21.5 years derived from least squares linear fits of seasonal-mean wind components from March 1979 to February 2022 to the average wind components for the period, shown for (a, b) DJF, (c, d) MAM, (e, f) JJA and (g, h) SON.**





Figure 11 shows weakening of the mean flow at high latitudes over Eurasia and north-western Canada in DJF, but
strengthening over Eurasia in MAM. More-focused plots for Eurasia confirm that the latitude of maximum mean flow shifts
southward in both seasons, consistent with the more evident southward shift over the eastern Atlantic and with the jet-stream
analysis of Manney and Hegglin (2018). The subtropical jet shifts northward in DJF but southward in MAM over the Middle
East and central Asia. Weakening of this jet in JJA has been discussed by Dong *et al*. (2022), who attribute it to
anthropogenic aerosol emissions.

The picture given here for the eastern North Pacific differs from that summarised by Manney and Hegglin (2018), who
concluded that the subtropical jet in this region has shifted strongly *southward* in DJF. The difference appears to stem from
Manney and Hegglin's characterization of the flow at a particular longitude as simply comprising up to one subtropical jet
and one polar jet. As the "subtropical" jet moves northward while crossing the North Pacific it reaches a point where it
becomes designated the "polar" jet, with the "subtropical" jet then designated to be the jet that crosses North America and
the North Atlantic, where it too eventually becomes a "polar" jet. Panels (a) and (b) of Fig. 11 show a pronounced *northward*
shift of the main jet over the North Pacific in DJF. The North American jet hardly changes its position over the Baja
California Peninsula, but does appear over time to originate further west and south. This is associated with the stronger
westerly flow in the upper-tropospheric Pacific branch of the Walker Circulation discussed earlier. What is seen in Manney
and Hegglin's study to be the southward movement of the subtropical jet thus appears not to be the movement of a physical
jet, but rather a change in the longitude at which the appellation "subtropical" changes from one jet to another.

### 4.4 Fits of ERA5 to wind observations

Additional confidence in these results stems from how closely and stably over time ERA5's background forecasts and
analyses fit the assimilated observations. This is illustrated here for the region of tropical easterlies, and the regions of
strongest extratropical northern-hemisphere flow.

Figure 12 shows monthly rates of wind observation for the 175 to 225 hPa layer, from 10° S to 10° N and 90° W to 150° E
where the easterly component predominates. The observations are those assimilated from radiosondes, aircraft and features
tracked by satellites. There is a net increase over time in the numbers of each type of observation, more so for the satellite
and aircraft data, although the latter show a drop in number early in 2020 when commercial traffic fell due to the COVID-19
pandemic.

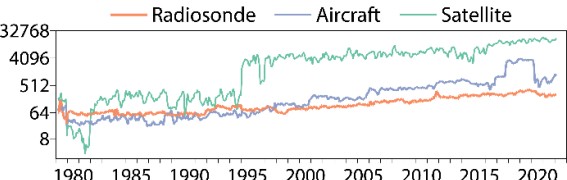


**Figure 12: Monthly average number of wind observations per day from 175 to 225 hPa, 10° S to 10° N and 90° W to 150° E, for
March 1979 to February 2022, from radiosondes (red), aircraft (blue) and satellites (green).**

Monthly means and standard deviations of the differences between these observations and the ERA5 background forecasts and analyses are shown in Fig. 13, for zonal and meridional wind components. The mean background and analysis
departures are especially close to zero for each type of observation from the late 1990s onwards. Earlier, the background zonal winds, and to a lesser extent the analysed winds, are a little higher than the observed winds. Interpretation of results such as these is not straightforward due to variations over time in data distributions, in possible observational biases and in the coverage of other types of observation that may have indirect influence on the analysed winds. What can be said, however, is that the trend in the analysed tropical easterlies at 200 hPa discussed earlier, whose magnitude exceeds 0.6
ms$^{-1}$/decade over the region in question, is much larger than any spurious contribution to the trend that Fig. 13 might suggest.

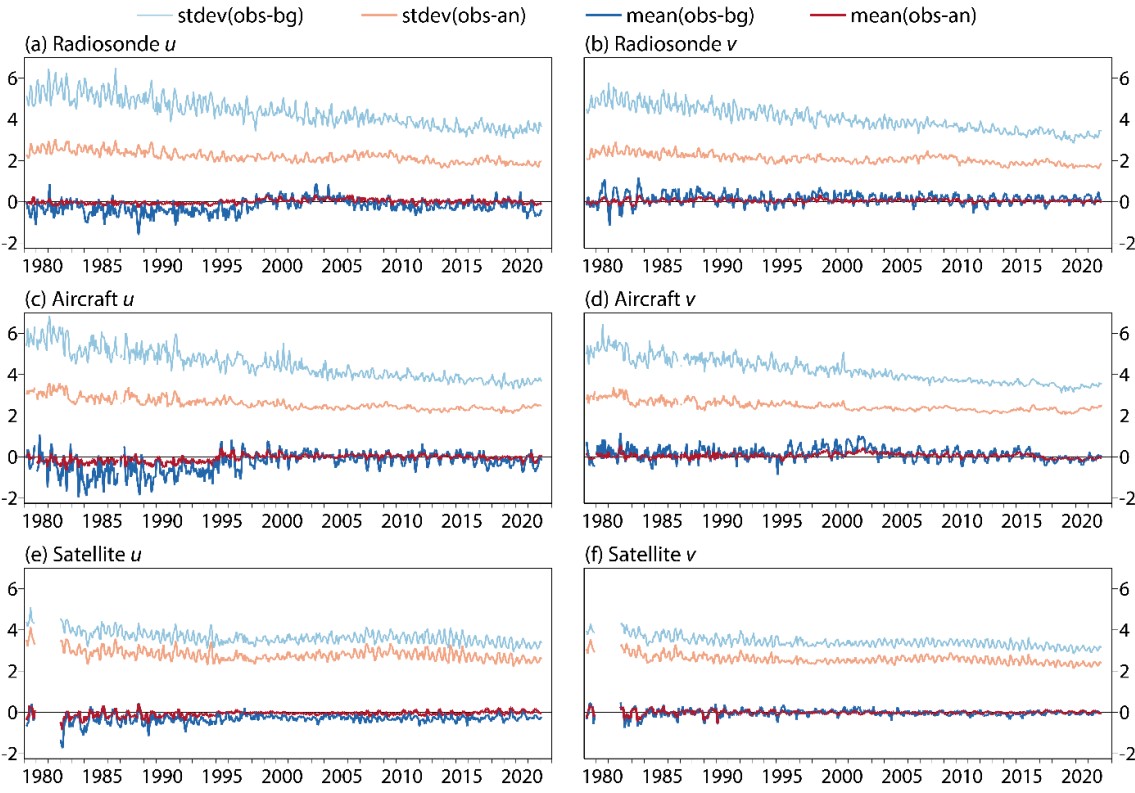

**Figure 13: Monthly means and standard deviations of observation minus background (dark/light blue) and observation minus analysis (red/orange) departures (ms$^{-1}$) for the 175 to 225 hPa layer over the region from 10° S to 10° N and 90° W to 150° E, from**
**March 1979 to February 2022, for zonal ($u$) and meridional ($v$) wind components from (a, b) radiosondes, (c, d) aircraft and (e, f) satellite imagery.**

The standard deviations show that the data assimilation system draws the analysis closer to the observations than background values are, throughout the period and for each type of observation. The decrease over time of the standard deviations of the background departures indicates improvements in the observing system over time. The analysis departures show smaller
improvement. They are constrained by the observation errors prescribed for the u and v wind components: 2.5 ms$^{-1}$ for



radiosonde data, 3.36 ms$^{-1}$ for aircraft data transmitted in AIREP code, 2.96 ms$^{-1}$ for other aircraft data and 5 ms$^{-1}$ for satellite-tracked winds, for the 200 hPa level.

Turning now to the northern extratropics, observation counts are shown in Fig. 14 for radiosonde data close to the eastern seaboard of North America and over eastern Asia, and for aircraft data further to the east over the western North Atlantic and

North Pacific, again for 175-225 hPa layer. Radiosonde numbers change little for most of the period, but increase in later years when soundings with higher vertical resolution become available. Aircraft numbers over the chosen region of the western North Atlantic increase substantially over time from the 1990s onward. This is not the case for the western North Pacific, although here too observation counts were at an all-time high immediately before the pandemic. Satellite-tracked winds are not shown for these regions as numbers are relatively low for the 175-225 hPa layer and have a pronounced

seasonal cycle.

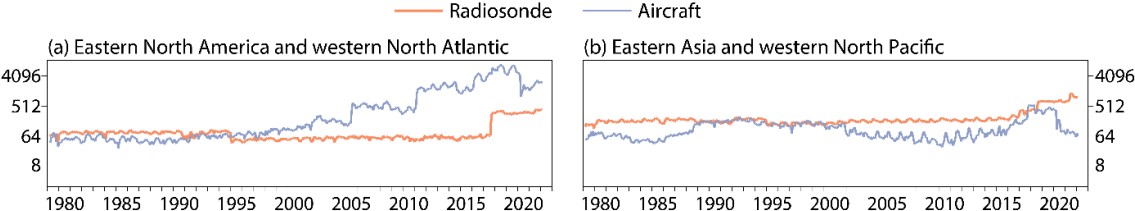

**Figure 14: Monthly average number of observations per day for the 175 to 225 hPa layer, from March 1979 to February 2022, for radiosondes (red) over the regions (a) from 35° N to 55° N and 80° W to 50° W and (b) from 25° N to 45° N and 115° E to 145° E and aircraft (blue) over the regions (a) from 35° N to 55° N and 60° W to 30° W and (b) from 25° N to 45° N and 145° E to 175° E.**

Corresponding data fits are presented in Fig. 15. Again, the mean fits of the analyses to the assimilated observations indicate no drift over time large enough to cast serious doubt on the nominal 43-year wind-speed changes of 1 ms$^{-1}$ or more derived for these jet-stream regions. Mean background and analysis departures from the radiosonde observations for North America do grow from around 2005 until these observations increase in number in 2017. The corresponding standard deviations of analysis departures hardly change over time for the zonal wind and increase over time for the meridional wind prior to 2017,

despite declining standard deviations of background departures. Departures from aircraft wind data for the North American region sampled for radiosonde data are much the same as those shown in Fig. 15 for the western Atlantic. The deterioration of the fit to North American radiosonde data is likely due to assimilation of increasing amounts of data on wind and temperature from various sources, and may in particular be a consequence of the issue with temperature biases in the lower troposphere noted in Sect. 3. Deterioration of the quality of the radiosonde data themselves is a less likely explanation. Only

a hint of similar behaviour can be seen for the Asia-Pacific region.

Fits to radiosonde and aircraft wind observations over all longitudes of the southern hemisphere from 10° S to 75° S have also been examined. Mean background departures are a little noisier for early years, but mean background and analysis departures in general remain close to zero throughout, behaving similarly to the mean fits shown in Fig. 15. Standard deviations are likewise similar towards the end of the period. The standard deviations of background departures are some 10-

20 % higher for early years, but are below 6 ms$^{-1}$ for all months. Analysis departures have standard deviations that decline





from at most 3 ms⁻¹ in early years to about 2 ms⁻¹. Mean wind departures computed separately for Australia and southern South America also give no indication of any drift over time in data fits. There is a paucity of upper-air wind observations over the southern oceans, but analysis systems such as used for ERA5 have access to several types of information there concerning surface winds, and thermal-wind information is provided by assimilation of satellite data.


**Figure 15: Monthly means and standard deviations of observation minus background (dark/light blue) and observation minus analysis (orange/red) departures (ms⁻¹) for the 175 to 225 hPa layer, from March 1979 to February 2022, for zonal (*u*) and meridional (*v*) wind components from radiosondes over the regions (a, c) 35° N to 55° N and 80° W to 50° W and (b, d) from 25° N to 45° N and 115° E to 145° E, and aircraft over the regions (e, g) from 35° N to 55° N and 60° W to 30° W and (f, h) from 25° N to 45° N and 145° E to 175° E.**




## 5 Jet-stream wind maxima

Changes over time in extreme jet-stream speeds have been investigated using time series of monthly maximum wind speeds computed over several domains from hourly ERA5 analyses. Time series of monthly means of corresponding daily maxima have also been examined. Figure 16 shows the series for the monthly maximum speeds of the full wind and its meridional
component over regions encompassing most of North America and the North Atlantic (120º W - 20º W; 20º N - 90º N) and Asia and the North Pacific (60º E - 120º W; 20º N - 90º N), and the southern hemisphere south of 20º S. Values of monthly anomalies relative to least-squares linear trends are plotted, as in Fig. 1.

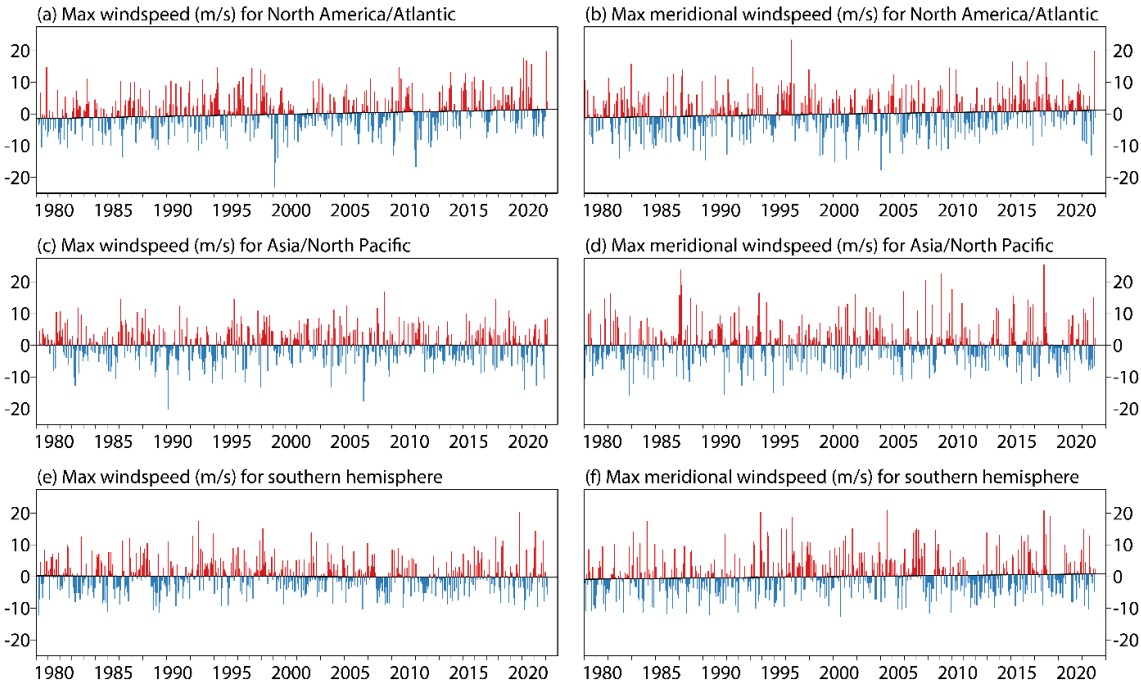

**Figure 16: Deviations from average values for each month in the period March 1979 to February 2022 in the highest 200 hPa**
**windspeed analysed during the month over regions covering most of (a) North America and the North Atlantic (120º W-20º W; 20º N-90º N), (c) Asia and the North Pacific (60º E-120º W; 20º N-90º N) and (e) the southern hemisphere south of 20º S. Panels (b), (d) and (f) are corresponding time series for the maximum speed of the meridional wind component. Values are plotted according to whether they are above (red) or below (blue) the least-squares-fit linear trend for the period.**

It is only for the North America/Atlantic region that there is a significant trend in monthly maxima of total wind speed. This
trend is for increasingly strong maxima at a rate of 0.67±0.48 ms⁻¹/decade. Rates are 0.00±0.40 ms⁻¹/decade for the Asia/North Pacific region and −0.11±0.37 ms⁻¹/decade for the southern hemisphere. The trends for individual jet-stream regions of the southern hemisphere are also not statistically significant.

The corresponding rate for the monthly maximum speed of the meridional wind component is 0.54±0.45 ms⁻¹/decade for the North America/North Atlantic region. That for the southern hemisphere is 0.39 ms⁻¹/decade with a 95 % confidence interval





of ±0.44 ms⁻¹/decade and a 90 % confidence interval ±0.37 ms⁻¹/decade. The trend for the Asia/North Pacific region is again far from significant.

The monthly means of daily maxima have additional significant trends. That for the North America/Atlantic region is 0.53±0.33 ms⁻¹/decade for total wind speed. The closest to matching it for significance is the trend for the polar jet eastward from 60⁰ E to 90⁰ W, south of 45⁰ S, which is 0.24 ms⁻¹/decade with a 95 % confidence interval of ±0.28 ms⁻¹/decade and

90 % confidence interval ±0.24 ms⁻¹/decade. Trends with 95 % confidence intervals for the meridional wind component are 0.37±0.22 ms⁻¹/decade for the North America/Atlantic region, 0.30±0.21 ms⁻¹/decade for the Asia/North Pacific region and 0.28±0.24 ms⁻¹/decade for the southern hemispheric polar jet.

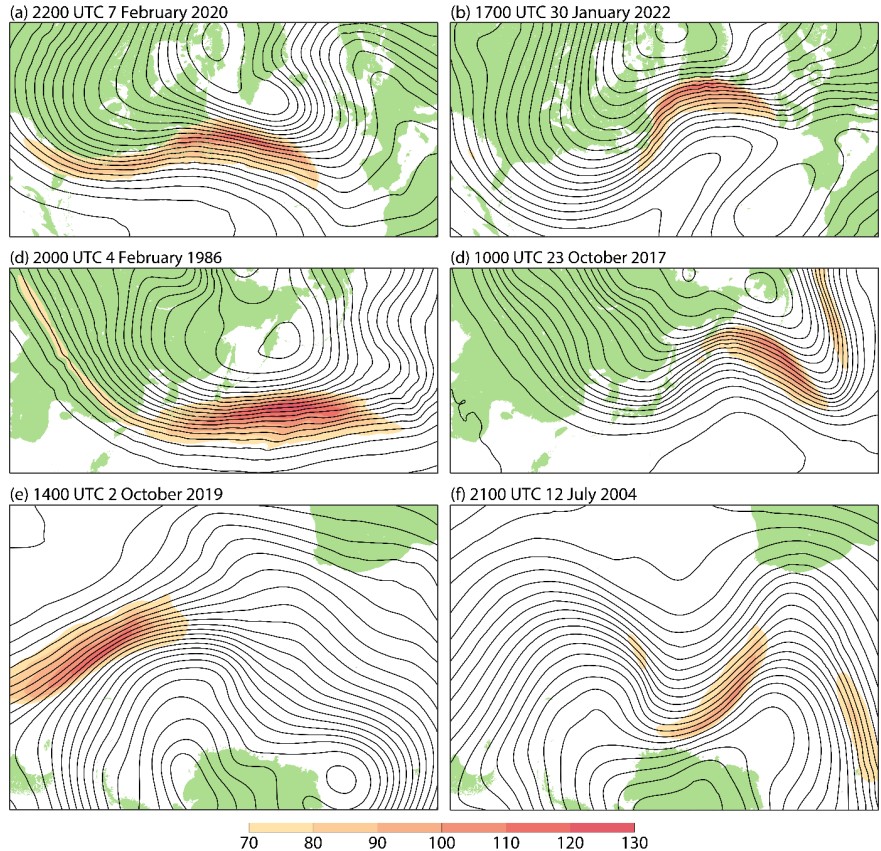

**Figure 17: 200 hPa geopotential height (contour interval 8dam) and windspeed (shading interval 10 ms⁻¹, for speeds ≥ 70 ms⁻¹),**
**showing North Atlantic jet streams for (a) 2200 UTC 7 February 2020 and (b) 1700 UTC 30 January 2022, North Pacific jet streams for (c) 2000 UTC 4 February 1986 and (d) 1000 UTC 23 October 2017 and South Atlantic jet streams for (e) 1400 UTC 2 October 2019 and (f) 2100 UTC 12 July 2004.**

Synoptic examples of extreme jet streams are presented in Fig. 17. Panel (a) shows a map for 2200 UTC on 7 February 2020, when the 200 hPa ERA5 wind speed reached its second highest value of all for the North America/Atlantic region. The case

is also that with the second most anomalously high wind speed relative to the monthly climatological average maximum



wind, shown by the second highest spike in Fig. 16(a). It is of special interest because 24 hours later the jet was aligned from south of Newfoundland to southern Ireland, with the region of strongest flow (the jet streak) moving north-eastward along the stream. This was particularly favourable for short eastward travel times for aircraft crossing the Atlantic that night. Record flight times were indeed reported[1]. At the surface a damaging named storm, Ciara, deepened to some 945 hPa west

of Scotland during the early hours of 9 February.

ERA5 winds on the second night reached 103 ms$^{-1}$ at the 200 hPa level and were stronger still lower down, with a peak speed of 121 ms$^{-1}$ at 272hPa (model level 81) at 0500 UTC on 9 February. The wind reports from aircraft assimilated in ERA5 are typically from levels higher than this, but they support a wind speed in excess of 100 ms$^{-1}$ around 200 hPa at this time. The strongest wind in aircraft reports is 108 ms$^{-1}$, from the 217 hPa flight level at 51.6° N, 24.9° W, for 0136 UTC on

9 February. The ERA5 analyses are consistent with the even stronger winds reported by aircraft for this level on the night of 7-8 February. 118 ms$^{-1}$ is the highest reported wind, from 217 hPa at 52.1° N, 36.7° W, for 2017 UTC on 7 February. The strongest 200 hPa wind from ERA5 for that night is 119 ms$^{-1}$.

Panels (c) and (e) of Fig. 17 show the cases of highest wind speed over the Asia/North Pacific region and the southern hemisphere south of 20º S. The maximum speeds are 126 ms$^{-1}$ over the North Pacific and 123 ms$^{-1}$ over the South Atlantic.

In terms of monthly anomaly, Fig. 16 shows that the Pacific case has the joint-second most anomalously high wind for the region, while the southern hemisphere case is the most anomalous for that hemisphere.

Panels (b), (d) and (f) show corresponding cases with highest speed of the meridional wind component. The Atlantic case involves strong poleward flow ahead of a trough over north-eastern Canada; the other two are associated with equatorward flow behind a sharp trough. Maximum speeds are 98 ms$^{-1}$ for the North Atlantic, 107 ms$^{-1}$ for the North Pacific and 99 ms$^{-1}$

for the southern hemisphere. The three cases are again either the most or second most anomalous in the data record.

The Atlantic case shown in panel (b) is also that with the highest total windspeed, 122 ms$^{-1}$ downstream of Greenland at 2200 UTC on the day shown. The reliability of this ERA5 value is more questionable because of the possibility of mis-represented orographic effects and the absence of confirmatory aircraft data. The highest reported wind close to 200 hPa assimilated by ERA5 on the day is 116 ms$^{-1}$. This was measured earlier, at 0931 UTC by an aircraft located at 197 hPa and

62.0° N, 40.0° W.

In each of the cases shown in Fig. 17, streaks of either negative (in the northern hemisphere) or positive (in the southern hemisphere) potential vorticity are found equatorward of the cores of the jet streams. This serves as a reminder that dynamical processes, such as inertial instability in this case (Thompson and Schultz, 2021), may place a limit on jet-stream strength notwithstanding any tendency for winds otherwise to increase.

---

[1]e.g. https://www.bbc.co.uk/news/uk-england-london-51433720; accessed 1 March 2022.



## 6 Lower-tropospheric wind changes

Wind changes in the lower troposphere have also been examined. Figure 18 shows the nominal 43-year changes in seasonal-mean wind speed as in Fig. 11, but for 850 hPa. Gulev et al. (2022) concluded in AR6 that most data products suggest increasing surface winds over the southern oceans, the western North Atlantic and the tropical eastern Pacific since 1980. This is the case also for the nominal 850 hPa wind changes shown here.

The predominant feature of the 850 hPa maps is the belt of strong westerlies around southern mid-latitudes. The nominal 43-year annual change is for a general strengthening of these winds, by 1.1 ms$^{-1}$ for the zonal-average wind speed where it is strongest. The annual strengthening is greatest over the eastern South Pacific, where the change reaches 1.9 ms$^{-1}$. Seasonally, the strengthening is greatest in DJF and MAM, when the increase in maximum zonal-average wind speed is 1.3 ms$^{-1}$. The latitude of this maximum shifts poleward by 2.2° in DJF, but changes little in other seasons.

The upper panels of Fig. 19 show time series of the monthly mean zonal wind component averaged over the southern extratropics between 45° S and 60° S. Westerly flow strengthens at a rate of 0.22±0.10 ms$^{-1}$/decade for ERA5 and 0.19±0.10 ms$^{-1}$/decade for JRA-55. The inter-monthly variability shown in these panels is similar for ERA5 and JRA-55, and principally associated with the SAM, known also as the Antarctic Oscillation, the leading mode of large-scale circulation variability in the southern extratropics on this timescale. Multi-decadal change in this region is commonly ascribed to a trend of the SAM towards stronger westerly flow, especially in austral summer when the flow maximum shifts poleward as well as strengthening. The "high confidence" AR6 statement of Gulev et al. (2022) noted in the Introduction is just one example of this. The alternative view that the multi-decadal change, represented by the linear trend in Fig. 19, should be regarded as change to the climatic state about which the SAM is defined, not to the SAM itself, is discussed briefly in Sect. 9.

Figure 18 shows clear increases in speed of the 850 hPa wind maximum over the western North Atlantic in DJF and MAM. The region of strongest flow over the eastern North Atlantic shifts southward except in SON. These changes at 850 hPa largely mirror those at 200 hPa. Weakening of the 850 hPa flow over the North Pacific is evident in all seasons other than JJA, with the DJF and MAM changes more prominent at 850 hPa than 200 hPa. As at 200 hPa, the region of strongest 850 hPa DJF flow over the eastern North Pacific shifts northward.

Easterly winds predominate at 850 hPa in the tropics over the Pacific and Atlantic oceans, and over the Indian Ocean south of the equator. The nominal 43-year changes from ERA5 are for strengthening flow over all these regions, in all four seasons. JRA-55 gives a more mixed picture for the tropical Atlantic, but otherwise is similar. Strengthening of the easterlies over a broad band of the Pacific is a well-established feature of the winds at 10m height from ERA-Interim (de Boisséson et al., 2014), and is especially robust statistically for the 850 hPa winds from ERA5 and JRA-55. Time series of the monthly mean zonal wind component averaged over tropical and subtropical latitudes of the Pacific Ocean are presented in panels (c) and (d) of Fig. 19. Trends are -0.26±0.11 ms$^{-1}$/decade for ERA5 and -0.24±0.10 ms$^{-1}$/decade for JRA-55.

Also prominent for JJA in Fig. 18 is the Somali Jet, for which ERA5 indicates a statistically significant increase in strength over time. However, although JRA-55 is in good agreement with ERA5 over most of the globe, it indicates a slight





weakening of this jet. A similar difference between the two reanalyses is found in the MAM and JJA averages of the southerly low-level jet located east of the Rockies in the southern USA. This too strengthens in ERA5 but not JRA-55. 675 Further assessment of these differences is required, but is beyond the scope of this study.

**Figure 18: Seasonal 850 hPa wind speeds (ms⁻¹) based on subtracting (left) and adding (right) the 21.5-year change derived from least squares linear fits of seasonal-mean wind components for 1979-2021 to the average wind components for the period, shown for (a, b) DJF, (c, d) MAM, (e, f) JJA and (g, h) SON. Regions where the ERA5 model orography exceeds 1500 m are masked in 680 black.**



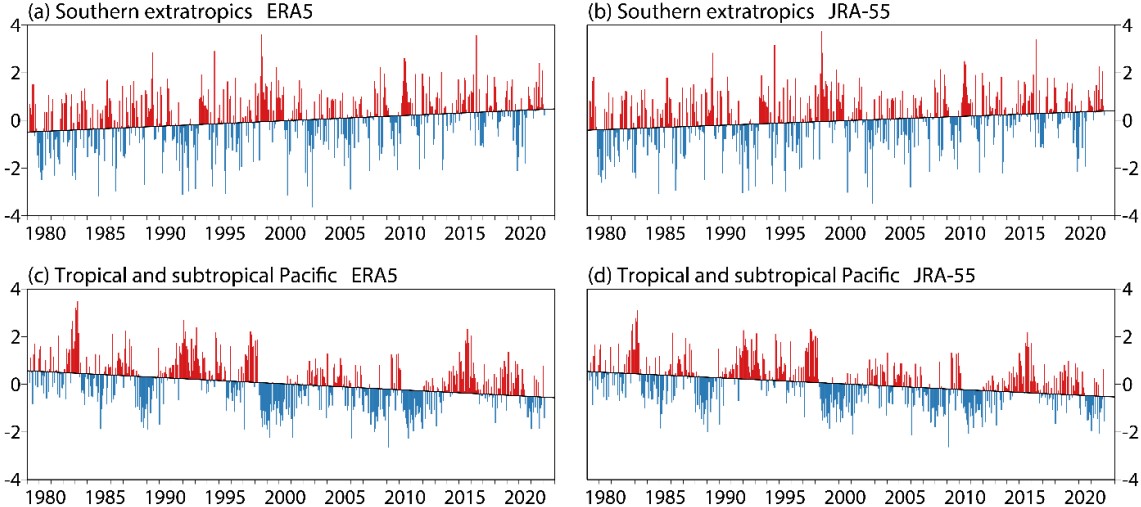

**Figure 19: Monthly means of 850 hPa zonal winds (ms⁻¹) averaged over the southern extratropics(45° S - 60° S) for (a) ERA5 and (b) JRA-55, and over central latitudes of the Pacific (30° N - 30° S, 160° E - 120° W) for (c) ERA5 and (d) JRA-55. They are shown relative to the average for each month in the period from March 1979 to February 2022, and plotted according to whether they are above (red) or below (blue) the least-squares-fit linear trend for this period.**

## 7 Geopotential height and surface pressure

An additional view of changes is provided by Fig. 20, which presents maps of the trends of 200, 500 and 700 hPa geopotential height and surface pressure. There is an increase in geopotential over most of the globe, where tropospheric temperature increases and surface pressure at most falls weakly. The geopotential increase is larger in the subtropics than the deep tropics in a band stretching westward from the Maritime Continent to South America, consistent with the strengthening of the tropical easterlies in this region discussed earlier.

The Antarctic is an exception, as there is not only slight cooling close to the coastline of Antarctica (Fig. 4) but also a decrease of surface pressure that additionally reduces geopotential on isobaric surfaces. Conversely, surface pressure tends to increase immediately to the north over the southern South Pacific, South Atlantic and Indian oceans, regions where temperature tends to increase. These changes are consistent with the increase in westerly wind over the southern oceans noted earlier. They are also characteristic of the changes seen over the life cycles of baroclinic waves (e.g. Simmons and Hoskins, 1978; Thorncroft et al., 1993), and thus suggestive of increased intensity or frequency of baroclinic disturbances. A trend for decreasing surface pressure also occurs over the Arctic Ocean, but not over land to the south. This partly counteracts the effect of arctic temperature amplification on the isobaric upper-tropospheric wind field.

Another feature of Fig. 20 is the wave train that extends eastward from the north-eastern Pacific Ocean to Asia. It implies a changing amplitude and phase of the stationary long-wave pattern. At the surface a trend towards higher pressure over the Pacific dominates, but the amplitudes of downstream troughs and ridges are more pronounced higher in the troposphere. The trend of the geostrophic zonal flow alternates between easterly and westerly south of the axis of the wave train, consistent

with the nominal decreases and increases in speed of the Asian subtropical jet shown in Table 3. The trend of decreasing
surface pressure between Greenland and western Europe as well as relatively weak warming aloft contributes to a 200 hPa
geopotential change consistent with the equatorward shift of the jet-stream region over the eastern North Atlantic. The
largest northern-hemispheric trends in the 200 hPa meridional wind component occur along the wave train. Meridional-wind
trends are also relatively large over the tropical and subtropical central South Pacific, where the subtropical jet weakens
downstream of Australia, and the time-mean flow strengthens in a band from the tropics to the South-American jet-stream
region (Figs. 8 and 9).

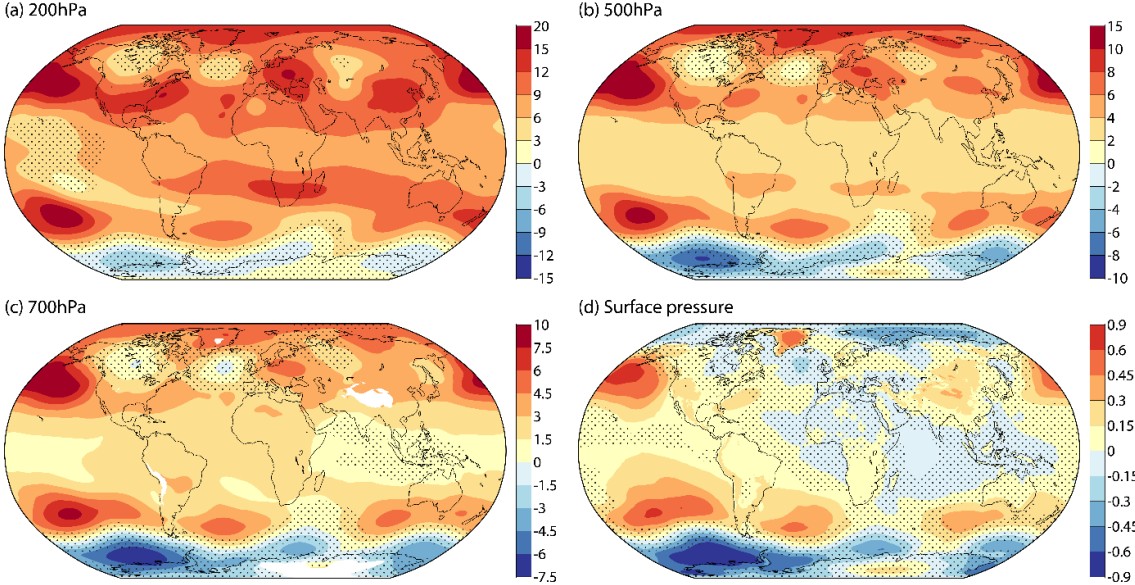

**Figure 20: Trends in geopotential height (m/decade) and surface pressure (hPa/decade) derived from least squares linear fits of
monthly mean deviations from average values for each month in the period March 1979 to February 2022. (a) 200 hPa height, (b)
500 hPa height, (c) 700 hPa height and (d) surface pressure. Dots show where the 95 % confidence interval is two-signed. 700 hPa
regions where the ERA5 model orography exceeds 3000 m are masked in white.**

The amplitude and wavelength of the northern hemispheric wave train varies with season. The wavelength is relatively short
in summer, consistent with expectations based on Rossby-wave behaviour (Hoskins and Woollings, 2015). The amplitude is
relatively large then, as can be seen from the 700 hPa and 300 hPa temperature maps for JJA in Fig. 5. In other seasons, the
wave train in upper-tropospheric geopotential reflect features predominant in the distribution of lower-tropospheric
temperature. In particular, DJF has a zonal wavenumber three pattern in geopotential that reflects trends for either cooling or
only relatively weak warming over western North America, the eastern Atlantic and western Europe, and Siberia, as shown
for surface-air temperature is Fig. 3 and upper-air temperature in Fig. 5. The trend towards higher surface pressure over the
north-eastern Pacific is most pronounced in winter. The austral summer sees the strongest trend towards lower surface
pressure around Antarctica, but the weakest trend towards higher pressure over the oceans immediately to the north.

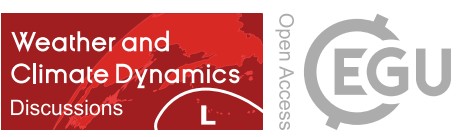

Statistical significance has not been found for trends in the sub-monthly contribution to the variances of geopotential and surface pressure computed for the extratropical hemispheres. This is in contrast with the trend for an increasing sub-monthly variation of kinetic energy density noted in the following section. The variances of geopotential and surface pressure decrease in polar regions, offsetting increases in middle latitudes. Geopotential variance over the region from 45° S to 60° S has an upward trend with one-signed 95% confidence interval at both 850hPa and 200hPa. This holds for the 90 % but not

the 95 % confidence level in the case of surface pressure. There is no such significant trend over northern mid-latitudes.

## 8 Kinetic and other forms of atmospheric energy

Trends in atmospheric energetics (Lorenz, 1967) provide additional measures of change in the general circulation. One of the products provided by ERA5 comprises geographically varying vertical integrals relating to several atmospheric budgets, including the energy budget. Annex 1 of the archive documentation for ERA-Interim (Berrisford et al., 2011) sets out the

735 equations and definitions used. Global integration provides values of components of the total energy of the atmosphere.

Figure 21 presents time series of monthly mean anomalies of (a) total energy and of its components:

(b) potential plus internal energy

$$\frac{1}{g}\sum(c_pT + \varphi_s)\,\Delta p$$

(c) latent energy

$$\frac{L}{g}\sum q\Delta p$$

and (e) kinetic energy

$$\frac{1}{g}\sum\frac{1}{2}(u^2 + v^2)\Delta p$$

where $T$ is temperature, $q$ is specific humidity, $u$ and $v$ are the horizontal wind components and $\varphi_s$ is surface geopotential. $g$ is the acceleration due to gravity, 9.80665 ms$^{-2}$, $L$ is the latent heat coefficient, taken as a constant 2.500810$^6$ Jkg$^{-1}$, and the

745 specific heat $c_p$ depends weakly on specific humidity:

$$c_p = c_{pd}(1 + \left(\frac{c_{pv}}{c_{pd}} - 1\right)q)$$

where $c_{pd}$=1004.709 Jkg$^{-1}$K$^{-1}$ and $c_{pv}$=1846.1 Jkg$^{-1}$K$^{-1}$. Variables have values for each of the ERA5 model's 137 layers, and the sums are taken over all layers. $\Delta p$ is the difference in pressure across each layer, which depends on surface pressure for pressures larger than about 75 hPa. Values are expressed as averages per unit area of the Earth's surface in units of 10$^7$ Jm$^{-2}$.

The potential plus internal energy is referred to as the total potential energy (TPE).

There is evident similarity between the variations over time of the total energy and of the global-average surface-air temperature (Fig. 1(a)). Total energy has the larger variation over time relative to the least squares trend, as seen for example





in the peaks associated with the 1997/98, 2009/10 and 2015/16 El Niño events. Its trend with 95 % confidence interval is $0.160\pm0.081$ $10^7$Jm$^{-2}$/decade.

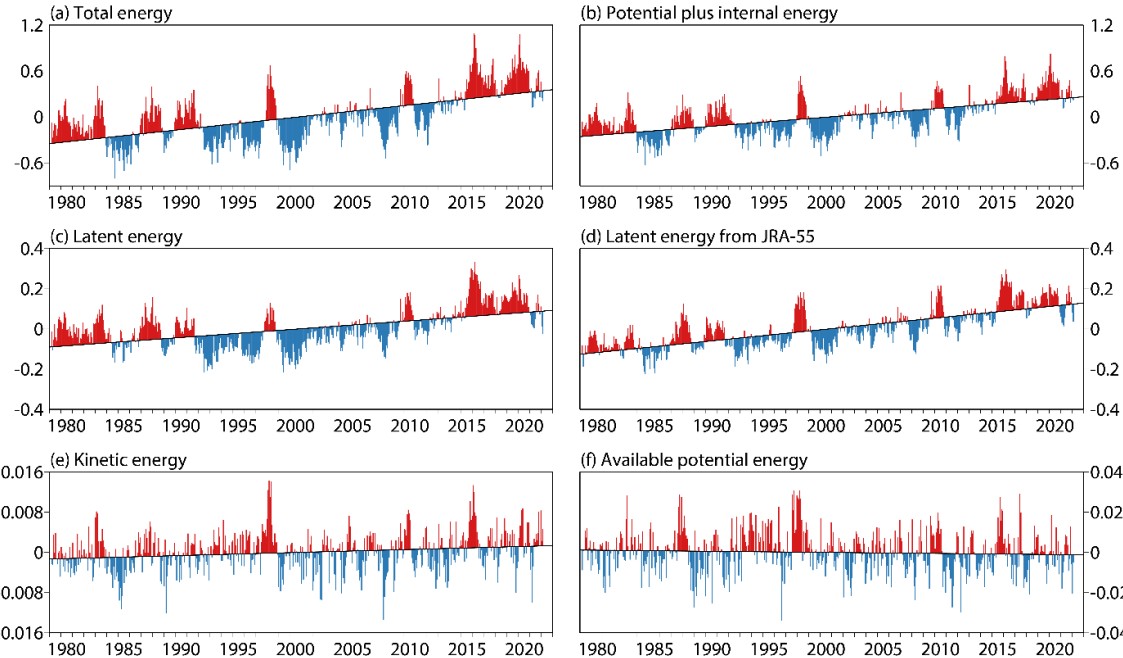

**Figure 21: Monthly mean ERA5 global (a) total energy, (b) potential plus internal energy (TPE), (c) latent energy, (e) kinetic energy and (f) available potential energy (APE), expressed as average values per unit area of the Earth's surface with unit $10^7$ J m$^{-2}$, relative to the average for each month in the period from March 1979 to February 2022. Panel (d) shows latent energy from JRA-55. Values are plotted according to whether they are above (red) or below (blue) the least-squares-fit linear trend for this period.**

The largest contributor to this trend is the $0.118\pm0.050$ $10^7$Jm$^{-2}$/decade trend in TPE. This is accounted for almost entirely by the trend of the thermal energy component:

$$\frac{1}{g}\sum c_p T \Delta p$$

As the dependence of the specific heat on moisture is relatively small, this is close to being directly proportional to the pressure-weighted average temperature of the atmosphere. A 1K temperature change throughout the atmosphere gives a change in dry thermal energy of 1.04 $10^7$Jm$^{-2}$, for a reference surface pressure of 1013.24 hPa. In reality, the stratosphere has cooled while the troposphere has warmed, so the thermal energy trend in units of $10^7$ Jm$^{-2}$/decade is numerically smaller than the tropospheric temperature trends in units of K/decade shown in Table 2.

Latent energy as defined with constant $L$ is directly proportional to the total column water vapour (TCWV). A change in TCWV of 1kgm$^{-2}$ changes latent energy by about 0.25 $10^7$Jm$^{-2}$. The trend in latent energy is smaller and more uncertain than that of TPE, at $0.042\pm0.025$ $10^7$Jm$^{-2}$/decade. Uncertainty arises not only from the monthly variability that influences the confidence interval, but also because of an apparently spurious shift in the ERA5 analyses. Panel (d) of Fig. 21 accordingly



shows latent energy derived using TCWV from JRA-55. In this case the trend in latent energy is higher in magnitude and less uncertain: $0.058\pm0.017$ $10^7$Jm$^{-2}$/decade. The higher trend is likely to be at least in part because of ERA5's shift between
1987 and 1991 to lower TCWV over the tropical oceans, due to assimilation of microwave imager data. This is seen in comparisons both with JRA-55 and with retrievals from SMMR and SSM/I data (Wentz and Francis, 1992; Wentz et al., 1997). In addition, it can be seen in Fig. 21 that it is the time series of latent energy from JRA-55 that is the closer in form to the time series of TPE, suggesting that TCWV variability from JRA-55 is more consistent with ERA5's temperature variability than is the TCWV variability from ERA5. Corresponding results from ERA-Interim are nevertheless poorer still
than these from ERA5. Hersbach et al. (2015, 2020) provide further information.

Trends in latent energy computed for the period 1991-2020 are $0.098\pm0.028$ $10^7$Jm$^{-2}$/decade for ERA5 and $0.083\pm0.028$ $10^7$ Jm$^{-2}$/decade for JRA-55. In this case it is JRA-55 that shifts relative to ERA5 and the SSM/I retrievals, to lower TCWV over the oceans around the year 2000. Both reanalyses nevertheless show a larger trend for 1991-2020 than for 1979-2021. The trend of TPE from ERA5 also increases, to $0.184\pm0.049$ $10^7$ Jm$^{-2}$/decade. As noted earlier, global-mean two-metre
temperature also has a larger trend for 1991-2020, but its relative increase over the 1979-2021 trend is smaller than seen for these components of atmospheric energy.

Kinetic energy accounts for only a small part of total atmospheric energy, but it too has a statistically significant upward trend: $0.00057\pm0.00046$ $10^7$Jm$^{-2}$/decade. Its time series shows pronounced peaks associated with El Niño events. Manney et al. (2021) have shown that although there are appreciable seasonal and regional variations in the relationships between jet-
stream speeds and El Niño events, subtropical jet speeds tend to be higher than normal during these events. Ma et al. (2021) found significant correlations between the sea-surface temperature fluctuations that characterize El Niño and La Niña and components of the atmospheric energy cycle derived from JRA-55.

Lorenz (1955) noted that TPE is not a good measure of the amount of energy available for conversion to kinetic energy under adiabatic flow. He defined an available potential energy (APE) of the atmosphere as the difference between the actual
TPE and the smallest value of TPE that could result from an adiabatic redistribution of mass. He furthermore derived an approximate expression for APE:

$$\frac{c_{pd}}{2g}\sum\frac{1}{\left(\bar{T}-\frac{c_{pd}}{R_d}\frac{p\Delta\bar{T}}{\Delta p}\right)}\overline{(T-\bar{T})^2}\Delta p$$

where the overbar denotes a global isobaric average and vertical discretization has been introduced. This form of APE has been evaluated from ERA5 data, forming contributions from the 850, 700, 600, 500, 400, 300, 250 and 200 hPa isobaric
levels and integrating from 1000 to 175 hPa, using $R_d$= 287.04 JK$^{-1}$kg$^{-1}$. It does not take water vapour into account, so cannot give an indication of the energy available to increase kinetic energy through latent-heat release. Following Lorenz' original derivation, the terms involving $\bar{T}$ do not include the time-averaging used in subsequent studies such as that of Ma et al. (2021) to enable separation of APE into stationary and transient components.





The time series of monthly mean anomalies of this form of APE is presented as panel (f) of Fig. 21. It shows that APE, like
KE, is relatively high during El Niño events. Lorenz (1955) argued that simultaneous increases in both components of
energy are indicative of a role of diabatic processes. Warming of the tropical atmosphere at times of El Niño increases the
temperature contrast between equator and pole, and thus APE, all else being equal. Aside from this variability, the downward
trend in APE from 1979 to 2021 is not statistically significant: $-0.00052\pm0.00118$ $10^7$ Jm$^{-2}$/decade.

Comparison with earlier studies is hampered by differences in formulation of the energy equations, in the reanalyses used, in
the length of the period studied and in data-processing details. The time series for kinetic energy shown here compares
reasonably well with that reported recently by Ma et al. (2021) for JRA-55. Agreement is less good for available potential
energy, for reasons that are unclear.

Time series of global averages of $0.5(u^2+v^2)$ from ERA5 split into monthly mean and sub-monthly variations have been
examined for the 850 hPa, 500 hPa and 200 hPa levels. The predominant upper-tropospheric contribution to the net kinetic
energy is dominated in magnitude by the monthly mean component, which is also the component for which there are
pronounced maxima during El Niño events. This component exhibits no significant long-term trend, however. The trend is
mainly from the sub-monthly variability, which increases at a rate of $1.05\pm0.45$ m$^2$s$^{-2}$/decade at 200 hPa, 55 % coming from
the $u^2$ component and 45 % from $v^2$. At 500 hPa it is again only the sub-monthly component that has a significant trend
$(0.21\pm0.18$ m$^2$s$^{-2}$/decade). Conversely, at 850 hPa there are significant trends in both the monthly mean $(0.36\pm0.14$
m$^2$s$^{-2}$/decade) and the sub-monthly component $(0.17\pm0.07$ m$^2$s$^{-2}$/decade). The $u^2$ and $v^2$ components contribute about
equally to the sub-monthly component at 500 hPa, but the $v^2$ component accounts for about 70 % of the change at 850 hPa.

The geographical distribution of the trends of $0.5(u^2+v^2)$ at 850 hPa is dominated by increasing monthly mean and sub-
monthly components along the band of strong westerlies around southern mid-latitudes. The monthly mean component at
200 hPa exhibits banded regions of increasing and decreasing kinetic energy density, reflecting the banded structure of
change in the zonal wind component shown in Fig. 7. These trends largely cancel in the global average. In contrast, the sub-
monthly component at 200 hPa predominantly increases, in several jet-stream regions and at high latitudes of the North
Pacific. It also increases over the tropical central Pacific, where enhanced westerlies facilitate inter-hemispheric Rossby-
wave propagation. More generally, Žagar et al. (2020) have shown increasing sub-seasonal variability of the most energetic
equatorial waves, for ERA5, JRA-55 and two earlier reanalyses.

**9 Discussion**

This examination of trends in tropospheric wind, temperature and other variables from ERA5 has identified changes since
1979 and indicated the confidence that can be placed in a number of the findings. Although it largely confirms results of
previous studies based on shorter periods and earlier reanalyses, it also identifies some changes that have received little
emphasis hitherto and others that have received rather more emphasis than hindsight appears to justify.





Among the latter, Arctic amplification can be linked with weakening westerly flow in some high-latitude regions and seasons, but the upper-level westerlies mainly strengthen at northern middle latitudes. Polar jets shift southward, and sub-monthly variability decreases at high latitudes. These results do not support the picture of a large-scale slowing and consequential increased meandering of the upper-tropospheric mid-latitude flow driven by Arctic amplification first proposed by Francis and Vavrus (2012). Sub-monthly variability does increase in middle latitudes, but can be influenced

there by factors other than or additional to Arctic amplification, such as increased moisture content and spatial variations in temperature change distant from the Arctic.

Related to this, a recent multi-modelling study of the impact on winter circulation of future Arctic sea-ice loss (Smith et al., 2022) shows weakened westerlies over a larger area than found using ERA5 and JRA-55, but generally smaller changes than seen in the reanalyses over the last four decades. This discrepancy between modelling and reanalysis suggests that factors

other than sea-ice loss have been important in changing the extratropical circulation over these decades, as neither model error (Smith et al., 2022) nor the differences between future and past sea-ice loss appears to provide an explanation.

The most pronounced trend to have received little earlier attention is the increasing strength and meridional extent of the tropical upper-tropospheric easterlies. Its depiction by ERA5 has strong statistical significance and is supported both by JRA-55 and by the closeness of fit of ERA5 to the upper-tropospheric wind observations it assimilates. The trend for

stronger easterlies occurs for all seasons, from the Maritime Continent westward across Africa and South America to the easternmost tropical Pacific. It is seen in zonal averages throughout the troposphere, though only weakly in the upper troposphere in DJF, when strengthening of the westerly upper-level flow of the Pacific cell of the Walker Circulation is strongest. It is linked with tropospheric warming that is larger in the subtropics and outer tropics than in the deep tropics.

This is a case for which changes over the past four decades are not necessarily indicative of the changes to be expected in

decades to come. For the Pacific Walker circulation, Chung et al. (2019) and Wu et al. (2021) have argued that the strengthening over recent decades has been due to natural variability and thus not in contradiction with the consensus climate-model projection of a weakening of the circulation under anthropogenic warming. Moreover, Huang et al. (2020) report a dramatic weakening of the upper-tropospheric easterly jet in recent projections using high-emission scenarios, which is also the converse of what ERA5 and JRA-55 show for the past forty years.

Gulev et al. (2022) noted in AR6 that a consistent poleward shift of the tropospheric extratropical jets since 1979 has been reported by studies using multiple reanalyses. This is the case here for the jet-stream region over the eastern North Pacific, which shifts northward in DJF. More generally, however, ERA5 supports the finding by Manney and Hegglin (2018) that the northern hemispheric polar jets shift mostly equatorward. Marked in the present results is a strengthening of the mean flow and equatorward shift of the jet-stream region over the eastern North Atlantic. This is seen in the lower troposphere as well

as at jet-stream level, and is linked through thermal-wind balance with the minimum in warming over the North Atlantic between Greenland and Europe, the so-called "warming hole". Among modelling studies of jet-stream changes in the region, Oudar et al. (2020) extended the work of Zappa and Shepherd (2017), examining potential drivers of change that included amplified warming of the tropical upper troposphere and the Arctic lower troposphere, strength of the winter polar



stratospheric vortex, and the warming hole. Multiple drivers of the warming hole itself have been discussed by Keil et al.
(2020).

The low-level westerlies in the extratropical southern hemisphere exhibit an increase in speed that is largest in DJF and MAM, with a poleward shift in DJF but little average latitudinal change in other seasons. The polar jet stream aloft strengthens in the Pacific sector. A number of other changes are quite pronounced for particular seasons, but tend to cancel to give mostly only small net shifts in position, many of them poleward. The largest shift in latitude is equatorward,
however. This is for the subtropical jet over and immediately downstream of South America. The subtropical jet that runs from the eastern Indian Ocean across Australia to the eastern Pacific weakens.

The change in the low-level southern hemispheric westerlies is commonly interpreted as the SAM (or Antarctic Oscillation) becoming increasingly in a positive phase. An alternative view suggested by Fig. 19 is that the multi-decadal change is primarily to the underlying climatic state not to a mode of variability of the circulation. Modelling results discussed in
several places in AR6 imply that the forced change, due mainly to increased greenhouse gases and stratospheric ozone depletion, resembles a particular phase of the SAM. It is accordingly seen in indices of the SAM that are based either on anomalies with respect to a fixed climate or on differences in surface pressure between latitudes. The change (even if reversible in the long term) is not however a free mode of variability or oscillation about a climatic state. This suggests that the SAM be defined as the leading empirical orthogonal function of anomalies in geopotential height or surface pressure
with respect to a slowly varying climatic state such as provided by a running 30-year mean or a linear trend. This SAM would change in character as the climatic state changes, but remain in essence modal. Something similar has been done by the Climate Prediction Center of the US National Weather Service in defining an "Oceanic Niño Index" in terms of deviations of sea-surface temperature not from a fixed climate but from multiple overlapping 30-year average values[2].

The general level of agreement between ERA5 and JRA-55 is reassuring, but some regional differences nevertheless require
further investigation. This is the case in particular for ERA5's underestimated temperature trend for the lower troposphere over North America. Temporal variations in ERA5's fit to upper-tropospheric radiosonde wind observations over the same continent also need attention. Both cases are likely to be due to mismatches between the assimilated bias-adjusted data from radiosondes and aircraft, and possibly data of other types. Neither reanalysis is expected to have a monopoly on correctness, however. Some issues may be resolved when JRA-55 is replaced by the new JRA-3Q reanalysis. Further investigation of
other issues should be undertaken as part of the preparations for ERA6, which is expected to begin production in 2024.

**Conflict of interest**

The author is a former manager of ECMWF's reanalysis programme and has advised on the production of ERA5. He has sought to make fair comparisons with the results of other reanalyses. He declares no other conflict of interest.

---

[2]https://origin.cpc.ncep.noaa.gov/products/analysis_monitoring/ensostuff/ONI_change.shtml; accessed 1 March 2022.



**Data and code availability**

This processing of ERA5 data made direct access to the primary data archive held at ECMWF, and used standard ECMWF software for data handling and graphics together with codes written for the specific purpose of this study. The latter were not designed for general use and such use cannot be supported. The ERA5 analyses can be downloaded from https://www.ecmwf.int. ECMWF is unable to provide public access to the archive of assimilated observations used in section 4.4 of this paper. JRA-55 data were originally downloaded from the Japan Meteorological Agency. Updates have 905 subsequently been downloaded monthly from https://diasjp.net. Version 4 GISTEMP data were downloaded on 15 March 2022 from https://data.giss.nasa.gov/gistemp/ and HadCRUT5.0.1.0 data were downloaded on 22 March 2022 from https://www.metoffice.gov.uk/hadobs/.

**Acknowledgments**

Comments on this work from colleagues are gratefully acknowledged. ERA5 relied on the work of many people, not only in 910 its production team but also among those at ECMWF and from outside who contributed to the development and refinement of the ECMWF forecast model and data assimilation system over the years, those from outside who developed the ancillary analyses of sea-surface temperature and sea-ice concentration and other ancillary datasets, and all who contributed to the reprocessed and additional sets of observations that were used. ECMWF implements the Copernicus Climate Change Service on behalf of the European Union, and ERA5 was produced with funding from this Service. A wider set of acknowledgments 915 is provided by Hersbach *et al.* (2020).

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
