# Peer review of "Trends in the tropospheric general circulation from 1979 to 2022"

_Weather and Climate Dynamics, 2022_

## Community Comment (CC1)

SC on "Trends in the tropospheric general circulation from 1979 to 2022" by Adrian Simmons
Comments from Gloria Manney & Michaela Hegglin

We read your paper with great interest and most enjoyed the thorough and comprehensive analysis, especially the lucid discussion of considerations in addition to confidence levels that should fold into interpretation of the robustness of trend results.

We find the discussion at the end of section 4.3 of differences in the present results from those of Manney and Hegglin (2018, hereinafter M&H) very interesting, and indeed the topic may be worthy of some future analysis of how the differences in methods fold into the results, and what we can learn from that about physical interpretation of those changes. We do have a few additional suggestions related to these differences in methods that might also fold into the appearance of different results, which you may want to consider and, if you feel it is appropriate, mention in your discussion of this point in the paper. Regardless of any changes you may make, we would look forward to further offline discussion of these points in the future.

One obvious difference, related to the point you emphasize about our definition of "subtropical" and "polar" jets, is that we are defining these jets at all whereas you are analyzing the full wind fields – particularly in regions / periods (such as the ones discussed in this paragraph) where the distributions of jets (aka windspeed maxima) is not well characterized by a single jet (or in some cases even just two jets), it is not clear to us that trends in the full wind fields would always necessarily be colocated with those in the windspeed maxima.

We also note that because the altitude of the jet cores (even within a given type) varies significantly in pressure at different times/locations (and indeed M&H found robust trends in both subtropical and polar jet core altitude), your method using a single pressure level could potentially "translate" a shift in jet core altitude/pressure as a weakening or strengthening of the jet.

Regarding our definition of "subtropical" vs "polar" jets, our identification of a jet as the "subtropical" jet is based not simply on latitude but on identifying the westerly jet across which a "tropopause break" occurs – thus it is a physically-based definition, and the transition between a "subtropical jet" at one longitude and a "polar jet" at another would necessarily require a change in this character. In this sense, we did not (for the analysis of the subtropical jets) show results that may "mix up" jet types because of simple geographic considerations.

On the other hand, we did make a choice in M&H to analyze only one "polar" jet at each longitude, that being the strongest jet poleward of the subtropical jet (or of 40 deg latitude if there is no subtropical jet). The region / season in question here is one where the complicated jet structure may not be well-described even by just **two** jets. Our jet identification method does identify and characterize up to five jets at each longitude in each hemisphere (after that we

identify just one of those as the subtropical jet and one as the polar jet), so future analysis accounting for more westerly jets, where present, could be illuminating.

Finally, a couple of suggestions on very minor details in the paper:

Re the discussion (re Fig. 7) around lines 345–356 regarding the results on the 2PVU surface (used here to represent the dynamical tropopause) in the tropics. In the examples we've looked at (including some of ERA5), the 2PVU surface is above (lower pressure than) 100hPa (which is near 380K, both being common levels used to represent the tropical tropopause) in much of the region between about ±10 deg latitude, so presumably at lower pressure than 200hPa in a substantially broader latitude region. The dynamical tropopause is, in fact, commonly taken to be 2PVU (or another PV value in that range with strong gradients) in the extratropics and 380K in the tropics (defined simply as wherever that PV value is above 380K). While this does not imply anything wrong with the discussion in this section of the paper, it does appear to complicate interpretation of the results, and it might be good to be a bit more explicit about where 2PVU does and does not represent the tropopause well.

It would be helpful, for those of us who are "geographically challenged", to give approximate latitude/longitude values for regions mentioned by name in the paper that might not be obvious to everyone on the maps (e.g., Barents Sea, line 47; Svalbard, line 317; Somali jet, line 671).

If it is easy to implement, it would be appreciated if the globe figures could include meridians and parallels to ease comparison of the study's results with other studies. It would also be helpful to label the color bars with the quantity and units they represent.

The caption for Fig. 10 seems a little confusing in how it describes the types of dashed lines that are overlaid, could this be reworded so that the two types of dashes are mentioned together so the reader doesn't start out wondering why the first dashed contour that meets their eye clearly doesn't match the value they are reading in the caption, e.g., something like "Short- and long-dashed contours in panels…show the 25 and 35 ms^-1 contours, respectively, from panels (a) and (c)"?

Figure 17, two panels are labeled (d) and none (c).

Line 699, "arctic" should be "Arctic".

Line 883, it would be clearer if you bounded "however" by commas.

---

## Author Comment (AC4)

***The comments from Reviewer 2 are shown below using a regular font; the author responses are in bold italics.***

This is an interesting paper which is more in the forma of a somewhat-descriptive survey rather than necessarily developing new physical insights. It can be seen as an update on, or complementary to, a few aspects of the recently-released IPCC/UNEP Working Group I AR6 report. It has some focus on the comparison of two modern reanalysis sets (and makes remarks as to where and, in some instances why, these reanalysis products differ.

The manuscript has potential to be an important contribution to the literature. However, it requires revision as outlined below.

***The reviewer is thanked for his assessment of the paper. I do though consider that the paper is more than "a somewhat-descriptive survey rather than necessarily developing new physical insights". The paper has identified through observations and their analysis some new facets of circulation change over the past forty years, and, as reviewer 1 puts it "questions some of the current narratives" which may be seen as a step on the road towards new physical insights.***

***I have made a number of revisions to the manuscript in relation to the specific comments of the reviewer. Where I have not done so I have offered an explanation below, and would be happy to discuss any of the points further, although I consider them all to be minor: no comment calls for a major revision of the text.***

As an overarching comment on the paper some of the explanations resented are based on the basic dynamics and thermo-dynamics of the atmosphere. For example, from line 36 comment is made … 'Changes in upper-tropospheric winds are linked to changes in surface flow and horizontal temperature gradients through the tendency of the atmosphere to remain close to thermal-wind balance. Where hydrostatic and geostrophic balance apply, the vertical shear of the wind is proportional to the temperature gradient across the direction of flow. The proportionality factor is larger at low than high latitudes'

Even though such statements are true, they don't really belong is a 'scientific' paper. In many cases these remarks are obvious and the implications will be clear to the reader. I strongly suggest culling such remarks; this will make for a better paper, and also makes a valuable contribution to reducing the length of quite a long submission.

***I do not consider it is such a bad thing to include a few remarks regarding the basic properties of the atmosphere, especially as early on as the second paragraph of a paper – one of the paragraphs that a non-specialist might dip into to try to ascertain "what's this all about then?" Moreover, as one goes further into the paper, some readers may be reassured that the data assimilation system producing an atmospheric reanalysis is getting the basics right in its processing of multiple types of observation with multiple types of error. And some specialists may welcome a reminder of a few basics that might not be at the forefront of their minds. As one tends to see temperature change discussed rather separately from circulation change, noting their inter-relationship does not seem amiss.***

***That said, I agree the quoted sentences are rather "over the top" and have shortened the paragraph concerned. I have also revised the opening of Sect. 4. I do not see anything more that cries out for change, but would be pleased to consider any further specific suggestions.***

Lines 30-32: Reinforce this message by referencing the more recent paper of Screen, Bracegirdle, and co-authors (2018), Polar climate change as manifest in atmospheric circulation, Curr. Clim. Change Reps., 4, 383-395, doi: 10.1007/s40641-018-0111-4.

*I have added a reference to the suggested paper. I have placed it lower down the opening paragraph, where circulation change in the southern hemisphere is now mentioned.  The reviewer is thanked for suggesting this reference.*

Lines 36-43: The remarks made a few lines earlier I the paper point to the compelc association and interactions between the thermodynamics and dynamics. In these introductory comments it would be very helpful to refer to the paper of

Theodore G. Shepherd, 2014: Atmospheric circulation as a source of uncertainty in climate change projections. Nature Geoscience, 7, 703-708, doi: 10.1038/ngeo2253

and his insights of '…

'nearly everything we have any confidence in when it comes to climate change is related to global patterns of surface temperature, which are primarily controlled by thermodynamics. In contrast, we have much less confidence in atmospheric circulation aspects of climate change, which are primarily controlled by dynamics and exert a strong control on regional climate'

 *I do not agree it would be helpful to include an explicit reference to Shepherd (2014), although the paper is one that contributes to AR6's timely discussion to which I refer. Shepherd's paper has a modelling focus rather than the observational focus of the present paper. I would argue that the confidence we have in surface temperature trends is primarily due not to an understanding of their control by thermodynamics, but to the relatively good observational coverage we have of air temperature over land and sea-surface temperature (to which marine air temperature is quite closely coupled), coverage that goes back to the 19th century. To quote AR4: "Warming of the climate system is unequivocal, as is now evident from* observations *[my emphasis] of increases in global average air and ocean temperatures, widespread melting of snow and ice, and rising global average sea level." The pattern of surface temperature change over sea depends on ocean circulation, even though the change is ultimately forced thermodynamically, as indeed is atmospheric circulation. And as soon as one gets to the sub-global scale at which circulation change occurs, the patterns of temperature change over land depend on both circulation dynamics and thermodynamic forcing. An implication of  thermal-wind balance is that the confidence we have in wind changes in the upper troposphere must be similar to the confidence we have in vertically averaged horizontal temperature gradients in the troposphere.*

Lines 92-94: For easy reading, it would be beneficial if these trend values and corresponding p values were presented as inserts into the four boxes within Figure 1.

*Trend values have been added to the figure. The reviewer is thanked for the suggestion.*

Line 100, caption of Figure 1: Even though it is perhaps obvious, it would be worth alerting the reader here that the y-axis ranges are different in the four time series. (This is mentioned in the text at lines 108-109, but should be emphasised here.)

*I prefer not to add this information to the caption. The differences in axes for the four panels has already made more evident by adding the trends to the figure (see above response), and as noted by the reviewer the point is also made in the text.*

Lines 117-120: Showing the (ERA5) trends that are significant (in part (b)) is interesting and helpful. It, of course, shows the same structure as in Figure 2a over the non-white areas. I appreciate that

the author is showing these two plots (with much common information) for clarity. However, I'm wondering if the same purpose could not be achieved by presenting these two in ONE part of the Figure. For example, some subtle stippling could be added to Figure 2a indicating where the trends are NOT significant. This could save showing one map, and perhaps would be easy to absorb the information.

*Stippling has been introduced, as employed already in Figures 3, 4, 5 and 20. This has been done for the JRA-55 trends as well as the ERA5 trends. The figure has been reduced to four panels by deleting the correlation map for JRA-55. Stippling does not provide a very clear plot in the case of Fig. 7, but in this case I have deleted the masked plots for the |PV|=2 and JRA-55 panels, as the masking is very similar to that shown for ERA5 at 200 hPa.*

Incidentally, I presume that by 'one-signed' the author means 'one-sided' – please use this more conventional terminology. Also, a justification is required by using such a test, given that there are regions of cooling over the globe. More appropriate to use the two-sided test?

*"one-signed" means one-signed not one-sided. The confidence intervals are two-sided, representing a range of values about the mean, limited both above and below, expressed as ± deviations from the mean. By one-signed I meant that all values in the confidence interval are either entirely positive or entirely negative. The introduction of stippling (see previous comment) has required some revision of the text and the phrase "one-signed" no longer appears. I have also spelt out that the confidence intervals are two-sided.*

Line 158: 'SST' has already been defined (at line 142).

*I thank the reviewer for spotting this. A correction has been made.*

Lines 181-182: On this S/N issue worthwhile to reference recent paper of

Luke J. Harrington, 2021: Temperature emergence at decision-relevant scales. Environmental Research Letters, 16, 094018, doi: 10.1088/1748-9326/ac19dc.

*I have read the paper concerned, and agree it is well worth referring to here. The reviewer is thanked for this suggestion. The text has been amended accordingly.*

Lines 209-211: This significant winter cooling over Eurasia and the northeast of the United States of America are important regional aspects of the complexity (and the consequences) of remote influences on the T2 trends. This warrants more attention than is presented here. Strongly suggest, for example, pointing out the role of teleconnections from the Arctic, high latitude blocking, Pacific SSTs etc. Making Reference to following will help on this:

Overland et al., 2019: Weakened potential vorticity barrier linked to recent winter Arctic sea ice loss and midlatitude cold extremes. J. Climate, 32, 4235-4261,

Luo, Xiao, and co-authors, 2016 - Impact of Ural blocking on winter Warm Arctic–Cold Eurasian anomalies. Part I: Blocking-induced amplification. J. Climate, 29, 3925-3947,

Dai, A. et al., 2020: 'Combined influences on North American winter air temperature variability from North Pacific blocking and the North Atlantic Oscillation: Subseasonal and interannual time scales'. J. Climate, **33**, 7101-7123, doi: 10.1175/JCLI-D-19-0327.1,

Rudeva, and coauthors, 2021. "Midlatitude winter extreme temperature events and connections with anomalies in the Arctic and tropics". J. Climate, **34**, 3733-3749.

*The winter cooling over Asia and spring cooling over Canada are important regional aspects. When writing the paper I was of the opinion that discussion of mechanisms such as mentioned by the reviewer was out of the scope of the paper, but included references to a small number of papers as evidence that there was ongoing work and debate on this. Clearly this approach was not very successful, so I have resolved the issue by removing these references and simply stating explicitly that discussion is beyond the scope of this [already long] paper.*

Lines 265-270: Paper should make clear the physical/dynamic reasons why the near-tropopause winds are of great relevance here. Also, to make clear in the text here that Figure 6 is associated with the 200 hPa wind – the reader is finally told this at line 295, which is a bit late.

*Changes have been made to the opening of Sect 4. This corrects an omission in the original paper by noting that there is little special relevance for showing the winds at 200hPa other than that the level is close to that where wind trends are largest in both the extratropics and the tropical upper troposphere, and is also a level where there are quite plentiful wind observations from aircraft and satellite feature-tracking, increasing confidence in the reanalyses there. I also note now that the extratropical (but not the tropical) wind changes are similar in pattern at 500hPa and to a lesser extent 850hPa, though smaller in magnitude. This avoids having to include what would have been a rather inconclusive discussion as to which level in the extratropical troposphere is the most significant dynamically.*

Lines 274-275: On this jet perspective consider citing the more recent works of …

Dong B, Sutton RT, Shaffrey L, Harvey B (2022) Recent decadal weakening of the summer Eurasian westerly jet attributable to anthropogenic aerosol emissions. Nature Comms. 13: 1148 doi: 10.1038/s41467-022-28816-5,

Hallam S, Josey SA, McCarthy GD, Hirschi JJM (2022) A regional (land–ocean) comparison of the seasonal to decadal variability of the Northern Hemisphere jet stream 1871–2011. Climate Dyn. doi: 10.1007/s00382-022-06185-5,

Liu X, Grise KM, Schmidt DF, Davis RE (2021) Regional Characteristics of Variability in the Northern Hemisphere Wintertime Polar Front Jet and Subtropical Jet in Observations and CMIP6 Models. J. Geophys. Res. 126: e2021JD034876 doi: 10.1029/2021JD034876.

*The Dong et al.(2022) paper is already referred to, albeit later in the paper. I do not think more is needed here.*

Line 301: The term 'variation' is used thru the manuscript to mean different things, from the qualitative concept and also to the mathematical variance. This can become a little confusing for the reader. Here the author is referring to '… the total sub-seasonal variation, the sum of the variances of the zonal and meridional wind'. This is clearly defined, but it should be given a more precise and informative name, such as 'summed variance'. In general use 'variance' when the statistical concept is being examined.

*I have changed the word "variance" to "variability" in two places. The "total variation" is the standard term for the trace of a variance/covariance matrix (see e.g. https://online.stat.psu.edu/stat505/book/export/html/645) , and in the present case of a horizontal-wind vector, the total variation is the sum of the variances of the zonal and meridional winds, which is what I wrote in the text aside from insertion of the word "sub-seasonal". I have deleted the latter word, as it is superfluous given the context.*

Lines 380-384: It would be helpful here (and other places where relevant) to relate these WC changes to the geographical distribution of tropical SST changes. This need not be comprehensive, and a few words will probably suffice. Worthwhile to not merely present a description of the changes, but also link them physically to other (driving?) processes.

*In section 8.3.2.3 of the AR6 WGI report, it is written "The causes of the observed strengthening of the Pacific WC during 1980-2014 are not well understood". Tropical SST changes are identified in the report as one factor, but other factors are also discussed. The results presented in this paper are relevant in that they increase confidence in the observational picture, but there is nothing I could add on driving processes that is beyond what is in AR6, and I already note in the paragraph in question that AR6 provides a discussion of the Pacific Walker Circulation. I do note in the concluding discussion that there is work indicating the strengthening is due to internal variability rather than forced climate change, and that the present results do not contradict this.*

Line 432: I am a bit confused by the use of the word 'nominal' here. Reference to Webster's did not help me greatly (e.g., 'existing or being something in name or form only', 'of, being, or relating to a designated or theoretical size that may vary from the actual', …). Please to use more conventional terminology here and below (using, e.g., 'de-trended', 'anomalies', …)

*I was not, and am still not, clear that there is a conventional terminology for what I decided to call the "nominal" climatic states at the beginning and end of the 43-year study period. "Nominal" came quickly to mind when I was thinking of a word, but before using it I did check several dictionaries. I have checked again. At https://www.merriam-webster.com/dictionary I find the definitions the reviewer quotes. But I also find other definitions there. Among them is "approximate". In my version of the Oxford English Dictionary one of the definitions is "existing in name only, not real or actual". An example of use in https://www.ldoceonline.com/dictionary is "A nominal diameter may also be derived from the volume of the pebble" which seems the equivalent of saying "a nominal climate for the start of the period may be derived from a later climate by use of a linear trend". I could have used "filtered" instead of "nominal" as I filter out variability by using the linear trend, but the word filtered is not specific and would also need to be defined in the present context. So I still prefer to use the word nominal and to define what I mean by the word. I have slightly revised the text at the beginning of Sect. 4.3 to emphasize more clearly that I am defining what I mean by "nominal", and I have checked that the word is used nowhere else in the paper other than with the strictly defined meaning.*

Line 686: Just confirming the paper is referring to 'surface pressure' rather than 'sea-level pressure'? If so, comment on why this choice was made.

*I confirm that when I wrote surface pressure I meant surface pressure not mean-sea-level pressure. This was to avoid either presenting unphysical extrapolated values under high orography, including Antarctica, or having to blank much larger parts of the map than are blanked for 700hPa geopotential. I do not consider it necessary to disrupt the flow of the text to explain this in the paper, as the reader should be able to infer this is the case as I have not applied the blanking that is used for extrapolated values of geopotential and (earlier in the paper) temperature. Moreover, the surface-pressure trends that are discussed in the text are all over sea, where the distinction hardly matters.*

Clearly space considerations have prevented the author from showing the SEASONAL trends in surface pressure, but the annual mean hides a lot of interesting seasonal behavior. To highlight this make reference in the paper to the study of Li et al. (2021 - Trends and variability in polar sea ice,

global atmospheric circulations and baroclinicity, Ann. NY Acad. Sci., 1504, 167-186, doi: 10.1111/nyas.14673) who show that the strong midlatitude N Pacific increases are dominated by the DJF trends, while the annual deepening in the Amundsen-Bellingshausen Seas are predominantly due to large reductions in the intermediate seasons.

***The figure changes noted above have freed space for an additional figure showing the DJF and JJA trends for 200hPa geopotential height and surface pressure. In addition I have included a reference to Simmonds and Li (2021) and thank the reviewer for drawing my attention to this work. As it was also based on ERA5, for a period that was only slightly different from that used in the present paper, I also looked at the corresponding trends from JRA-55. They are reassuringly similar, differing appreciably only over the Antarctic Plateau. I have added a sentence on this.***

---

## Author Comment (AC5)

We read your paper with great interest and most enjoyed the thorough and comprehensive analysis, especially the lucid discussion of considerations in addition to confidence levels that should fold into interpretation of the robustness of trend results.

We find the discussion at the end of section 4.3 of differences in the present results from those of Manney and Hegglin (2018, hereinafter M&H) very interesting, and indeed the topic may be worthy of some future analysis of how the differences in methods fold into the results, and what we can learn from that about physical interpretation of those changes. We do have a few additional suggestions related to these differences in methods that might also fold into the appearance of different results, which you may want to consider and, if you feel it is appropriate, mention in your discussion of this point in the paper. Regardless of any changes you may make, we would look forward to further offline discussion of these points in the future.

***I am grateful for this comment, and open to further discussion and possible additional future analysis. I have made some changes to the text, as indicated below.***

One obvious difference, related to the point you emphasize about our definition of "subtropical" and "polar" jets, is that we are defining these jets at all whereas you are analyzing the full wind fields – particularly in regions / periods (such as the ones discussed in this paragraph) where the distributions of jets (aka windspeed maxima) is not well characterized by a single jet (or in some cases even just two jets), it is not clear to us that trends in the full wind fields would always necessarily be colocated with those in the windspeed maxima.

***I have expanded the initial discussion of the differences between our approaches in Sect. 4.1 to include a little more on M&H's identification of subtropical and polar jet streams, including jet-core heights and their changes. I already note in the context of Fig.10 that changes in the monthly mean wind speed are qualitatively similar to changes in the speed of the monthly mean wind, notwithstanding the former being larger in magnitude than the latter.***

We also note that because the altitude of the jet cores (even within a given type) varies significantly in pressure at different times/locations (and indeed M&H found robust trends in both subtropical and polar jet core altitude), your method using a single pressure level could potentially "translate" a shift in jet core altitude/pressure as a weakening or strengthening of the jet.

***Yes, but only to a small degree. In responding to a point made by reviewer 2 I have noted that the extratropical trends in the speed of monthly mean winds have much the same structure at 500hPa as at 200hPa, though smaller magnitude. Something of the same can be said for 850hPa, as already noted in the paper. The changes at 200hPa are related to the changes in temperature gradient lower in the troposphere through the thermal wind relationship, and the similarities with the changes lower in the troposphere suggest that changes that result from changes in tropopause height or structure can be only a small fraction of the net 200hPa wind changes. The similarity between mid-latitude wind trends at 200hPa and |PV|=2 bear this out. The point does need to be made that I am not presenting the changes in peak jet speed, however. I have accordingly revised text in Sect. 4.2.***

Regarding our definition of "subtropical" vs "polar" jets, our identification of a jet as the "subtropical" jet is based not simply on latitude but on identifying the westerly jet across which a "tropopause break" occurs – thus it is a physically-based definition, and the transition between a "subtropical jet" at one longitude and a "polar jet" at another would necessarily require a change in this character. In this sense, we did not (for the analysis of the subtropical jets) show results that may "mix up" jet types because of simple geographic considerations.

On the other hand, we did make a choice in M&H to analyze only one "polar" jet at each longitude, that being the strongest jet poleward of the subtropical jet (or of 40 deg latitude if there is no subtropical jet). The region / season in question here is one where the complicated jet structure may

not be well-described even by just two jets. Our jet identification method does identify and characterize up to five jets at each longitude in each hemisphere (after that we identify just one of those as the subtropical jet and one as the polar jet), so future analysis accounting for more westerly jets, where present, could be illuminating.

*I believe the changes to Sect. 4.1 and Sect. 4.2 noted above are almost all that is needed. I have made only small revisions to the final paragraph of Sect. 4.3.*

Finally, a couple of suggestions on very minor details in the paper:

Re the discussion (re Fig. 7) around lines 345–356 regarding the results on the 2PVU surface (used here to represent the dynamical tropopause) in the tropics. In the examples we've looked at (including some of ERA5), the 2PVU surface is above (lower pressure than) 100hPa (which is near 380K, both being common levels used to represent the tropical tropopause) in much of the region between about ±10 deg latitude, so presumably at lower pressure than 200hPa in a substantially broader latitude region. The dynamical tropopause is, in fact, commonly taken to be 2PVU (or another PV value in that range with strong gradients) in the extratropics and 380K in the tropics (defined simply as wherever that PV value is above 380K). While this does not imply anything wrong with the discussion in this section of the paper, it does appear to complicate interpretation of the results, and it might be good to be a bit more explicit about where 2PVU does and does not represent the tropopause well.

*Reviewer 1 made a related comment, and I have amended the text to note that the pressure to which winds are interpolated, though first evaluated using a calculated value of PV, is not allowed to be lower than 89hPa. What I did not include, but can be found in the public documentation to which Hersbach et al. (2020) refer, is that all fields are fitted with spherical harmonics before archiving, so that pressures retrieved from the archive can have local (small-scale) values less than 89hPa. But these low values were not used in the interpolation of fields to the nominal |PV|=2 surface.*

It would be helpful, for those of us who are "geographically challenged", to give approximate latitude/longitude values for regions mentioned by name in the paper that might not be obvious to everyone on the maps (e.g., Barents Sea, line 47; Svalbard, line 317; Somali jet, line 671).

*I have made some minor re-wordings which may help. A balance has to be struck in matters such as this, as it is easy to irritate those who know well what one is talking about – as I once found out after giving a talk where I deliberately avoided referring to the Barents Sea by name. For the geographically challenged, which includes all of us to some extent, Google provides a quick way of finding what one wants, and helps in the process of becoming less-challenged.*

If it is easy to implement, it would be appreciated if the globe figures could include meridians and parallels to ease comparison of the study's results with other studies. It would also be helpful to label the color bars with the quantity and units they represent.

*I prefer not to make these changes. Adding latitude and longitude lines would inevitably make it a little more difficult to see the detail displayed in the maps, particularly near the poles. The maps include coastlines and discussion in the text tends to refer to placenames rather than latitude and longitude, which makes for an easier read for those who are not geographically challenged (see above response). As to adding units to all the colour bars, I can of course do this quite quickly, and will do so if an editor requests it. But adding  units would cause the maps to be smaller for a given figure dimension. In most if not all cases the unit is stated in the first line of the figure caption, so should be picked up quickly by the viewer.*

The caption for Fig. 10 seems a little confusing in how it describes the types of dashed lines that are overlaid, could this be reworded so that the two types of dashes are mentioned together so the

reader doesn't start out wondering why the first dashed contour that meets their eye clearly doesn't match the value they are reading in the caption, e.g., something like "Short- and long-dashed contours in panels…show the 25 and 35 ms^-1 contours, respectively, from panels (a) and (c)"?

***The caption has been revised. The result is shorter and I hope clearer. Thanks for pointing this out.***

Figure 17, two panels are labeled (d) and none (c).

***Thanks for spotting this. The correction has been made.***

Line 699, "arctic" should be "Arctic".

***Thanks also for spotting this one. The correction has been made.***

Line 883, it would be clearer if you bounded "however" by commas

***I have made this change too.***